# CMTM3 regulates neutrophil activation and aggravates sepsis through TLR4 signaling

Haiyan Xue [1,2,3], Ziyan Xiao [1], Xiujuan Zhao[1], Shu Li[1], Qian Cheng[3], Chun Fu[1] & Fengxue Zhu [1,2]✉

## Abstract

**Regulation of neutrophil activation plays a significant role in managing sepsis. CKLF-like MARVEL transmembrane domain containing (CMTM)3 is a membrane protein involved in immune response. Here, we find that CMTM3 expression is elevated in sepsis and plays a crucial role in mediating the imbalance of neutrophil migration. *Cmtm3* knockout improves the survival rate of septic mice, mitigate inflammatory responses, and ameliorate organ damage. Mechanistically, the deletion of Cmtm3 reduced the expression of Toll-like receptor 4 (TLR4) on neutrophils, leading to a decrease in the expression of C-X-C motif chemokine receptor 2 (CXCR2) on the cell membrane. This resulted in a reduced migration of neutrophils from the bone marrow to the bloodstream, thereby attenuating their recruitment to vital organs. Our findings suggest that targeting CMTM3 holds promise as a therapeutic approach to ameliorate the dysregulation of neutrophil migration and multi-organ damage associated with sepsis.**

**Keywords** Sepsis; CMTM3; Neutrophil; Inflammation
**Subject Categories** Immunology; Molecular Biology of Disease; Signal Transduction

## Introduction

Sepsis, a life-threatening condition characterized by multi-organ dysfunction, arises primarily from a dysregulated host response to infection (Singer et al, 2016). It stands as a leading cause of mortality in intensive care units, yet currently lacks approved targeted therapeutics (Rhee et al, 2019; Rudd et al, 2020). Treatment modalities for sepsis primarily encompass antibiotic administration, respiratory support, fluid therapy, and organ function support (Evans et al, 2021). The intricate relationship between the immune status and the pathogenesis, progression, and prognosis of sepsis has prompted significant research interest in immunomodulatory interventions (Hotchkiss et al, 2013; Wiersinga and van der Poll, 2022; Yao et al, 2020). As such, immunomodulatory therapy has emerged as a prominent area of investigation in the field.

The dysregulated immune response is considered a critical determinant in distinguishing sepsis from infection, with neutrophils emerging as potential candidates that warrant further investigation (Margraf et al, 2022; Shen et al, 2021). Under physiological conditions, the synthesis, mobilization, and subsequent recycling of bone marrow (BM)-derived neutrophils are tightly regulated processes that achieve a dynamic equilibrium, thereby ensuring the maintenance of a constant level of circulating neutrophils (Hidalgo et al, 2019). In the context of sepsis, there is frequently a pronounced and swift escalation in the number of neutrophils within the peripheral blood (Zhou and Sun, 2022). This surge in neutrophil release into the peripheral circulation can precipitate a diminution of the bone marrow's neutrophil reservoir, culminating in the mobilization of immature cells into the bloodstream (Shen et al, 2017). These immature neutrophils exhibit compromised recognition and phagocytic capabilities, rendering them ineffective in pathogen clearance. In addition, their reduced deformability predisposes them to aggregation within capillaries, leading to vascular blockages, tissue hypoxia, and subsequent organ damage (Huang et al, 2019; Kolaczkowska and Kubes, 2013; Sonego et al, 2016). Consequently, the precise modulation of neutrophil migration and release is imperative for the restoration and maintenance of immune equilibrium.

CMTM3 is a member of the CKLF-like MARVEL transmembrane domain gene superfamily, which is widely expressed in the immune system and plays a crucial role in immune response and tumor development (Han et al, 2003; Shen et al, 2022; Wang et al, 2009). Previous studies by Professor Han et al have found that CMTM3 is localized in early endosomes and enhances Rab5 activity to promote epidermal growth factor receptor (EGFR) internalization and degradation (Yuan et al, 2017). Ihsan Chrifi et al have demonstrated that CMTM3 can regulate angiogenesis by promoting VE-cadherin internalization and degradation (Chrifi et al, 2017). Imamura Y et al have shown that CMTM3 is a chaperone of the B-cell linker protein (BLNK), linking the B-cell receptor (BCR) and activating BLNK-mediated signaling pathway (Imamura et al, 2004). However, the precise role of CMTM3 in sepsis remains elusive, warranting further investigation. Therefore, exploring the precise function and mechanism of CMTM3 in sepsis is of great significance.

Here, through comprehensive bioinformatics analysis, we have discovered a substantial upregulation of *CMTM3* expression in septic patients. Consequently, we employed a functional loss-of-function strategy, utilizing a systemic *Cmtm3* knockout (KO) mice

[1]Department of Critical Care Medicine, Peking University People's Hospital, Beijing, China. [2]National Center for Trauma Medicine of China, Beijing, China. [3]Beijing Key Surgical Basic Research Laboratory of Liver Cirrhosis and Liver Cancer, Peking University People's Hospital, Beijing, China. ✉E-mail: Fengxue_Zhu@126.com

**Table 1. The characteristics of the datasets and samples used in this study.**

| Dataset ID | Sepsis | Healthy | Survival | Death | Platform |
|---|---|---|---|---|---|
| GSE33118 | 20 | 0 | 10 | 10 | GPL570 [HG-U133_Plus_2] Affymetrix Human Genome U133 Plus 2.0 Array |
| GSE54514 | 35 | 18 | 26 | 9 | GPL6947 Illumina HumanHT-12 V3.0 expression beadchip |
| GSE95233 | 51 | 22 | 34 | 17 | GPL570 [HG-U133_Plus_2] Affymetrix Human Genome U133 Plus 2.0 Array |
| Total | 106 | 40 | 70 | 36 | |

model, to elucidate the role of CMTM3 in the context of sepsis. Remarkably, our investigation revealed that *Cmtm3* deficiency resulted in a remarkable amelioration of sepsis survival rates, attenuation of inflammatory responses, and mitigation of tissue and organ damage in septic mice. Mechanistic studies revealed that knocking out *Cmtm3* reduced the expression of TLR4 on neutrophils, subsequently leading to a decrease in the membrane expression of CXCR2. This ultimately results in a reduced migration of neutrophils from the bone marrow to the bloodstream, thereby diminishing their recruitment to vital organs. Our research findings suggest that targeting CMTM3 could serve as a target to improve the imbalance of neutrophil migration in sepsis immunotherapy.

## Results

### CMTM3 is upregulated in sepsis patients and is positively correlated with inflammatory responses

Upon careful evaluation, the GSE33118, GSE54514, and GSE95233 datasets, which collectively include peripheral whole blood cells from a total of 106 sepsis patients and 40 healthy individuals, were selected for inclusion in our analysis (Parnell et al, 2013; Tabone et al, 2018; Venet et al, 2017; Data ref: Bilbault et al, 2017; Data ref: Pachot et al, 2017; Data ref: Parnell and Tang, 2014). Detailed information regarding the datasets can be found in Table 1. Subsequently, we employed the SVA package to address inter-batch differences. As shown in Fig. 1A, the presence of batch effects noticeably impacted the clustering of samples. However, after applying SVA, the batch effects were alleviated, as depicted in Fig. 1B.

We conducted an analysis of *CMTM1-8* expression and generated a heatmap to visualize the results, as illustrated in Fig. 1C. Comparing the expression levels of *CMTM1-8* between sepsis patients and healthy individuals, as presented in Fig. 1D, we observed that the expression levels of CMTM1-6 were consistently higher in sepsis patients compared to healthy individuals. Moreover, we performed a comparative analysis of expression levels between survival and non-survival sepsis patients, and the results shown in Fig. 1E revealed a significant difference only in the expression of *CMTM3*.

Macrophages and neutrophils play crucial roles in the inflammatory response during sepsis, secreting inflammatory mediators such as TNF-α and IL-1β to enhance the inflammatory response. Our analysis of immune cell infiltration types revealed that neutrophils and macrophages, which promote inflammation, were elevated in sepsis patients compared to healthy individuals, while Natural killer (NK) cells and CD8 cells were lower (Fig. EV1A). Further investigation of the correlation between *CMTM3* and inflammatory cells and mediators exhibited a significant positive correlation between *CMTM3* and neutrophils, macrophages, *TNF*, and *IL1B* (Fig. 1F–I). Furthermore, Fig. EV1B demonstrates the correlation between *CMTM1-8* and immune cells, while Fig. EV1C shows the correlation between *CMTM1-8* and inflammatory factors.

### Knockout of *Cmtm3* improves the survival rate and reduces inflammatory response in CLP mice

In order to investigate the function of CMTM3 in sepsis, we conducted further research using *Cmtm3* KO mice. The construction strategy and genotyping results of the mice are presented in Fig. EV2A,B. We established the CLP sepsis model in both wild-type (WT) and knoukout (KO) mice and collected serum samples 24 h later to measure the expression of inflammatory factors and organ injury markers. The results demonstrated that *Cmtm3* KO reduced the release of TNF-α, IL-1β, and IL-10 in the peripheral blood of CLP mice (Fig. 2A–C). In addition, the expression of AST, ALT, and sCr in the peripheral blood of KO mice was lower compared to WT mice in the CLP model (Fig. 2D–F). To assess histopathological damage, we examined liver, lung, and kidney tissues through HE stains. The results, as shown in Fig. 2G, illustrated that *Cmtm3* KO alleviated the pathological injury in the liver, lungs, and kidneys of CLP mice. Figure 2H–J provide the quantitative scoring of the injury severity in these organs, respectively. Importantly, we monitored the survival rates of WT and KO mice (15 pairs) for 7 consecutive days following CLP surgery. The survival analysis revealed that *Cmtm3* KO significantly improved the survival rate of CLP mice (Fig. 2K).

### Knockout of *Cmtm3* affects the release and retention of BM neutrophils in CLP mice

To investigate the role of CMTM3 in sepsis, we extracted the top 500 genes showing significant correlation with *CMTM3* expression for further analysis. The results of the GO - biological process (BP) enrichment analysis revealed that these genes were primarily associated with neutrophil activation and migration (Fig. 3A). Subsequently, we examined the impact of *Cmtm3* KO on neutrophil migration in the CLP model. We performed IHC staining of Ly6G to elucidate neutrophil infiltration in the tissues, and the results exhibited a lower neutrophil count in KO mice compared to WT mice in the CLP model (Fig. 3B). Consistent with the tissue's findings, the distribution of neutrophils in the peripheral blood demonstrated a similar pattern. As depicted in Fig. 3C,D, the population of peripheral blood neutrophil was significantly lower in *Cmtm3* KO mice compared to WT mice in the CLP model. Conversely, KO mice exhibited a higher BM neutrophil count compared to WT mice in the CLP model (Fig. 3E,F). These results suggested that knocking out *Cmtm3* ameliorate the excessive release

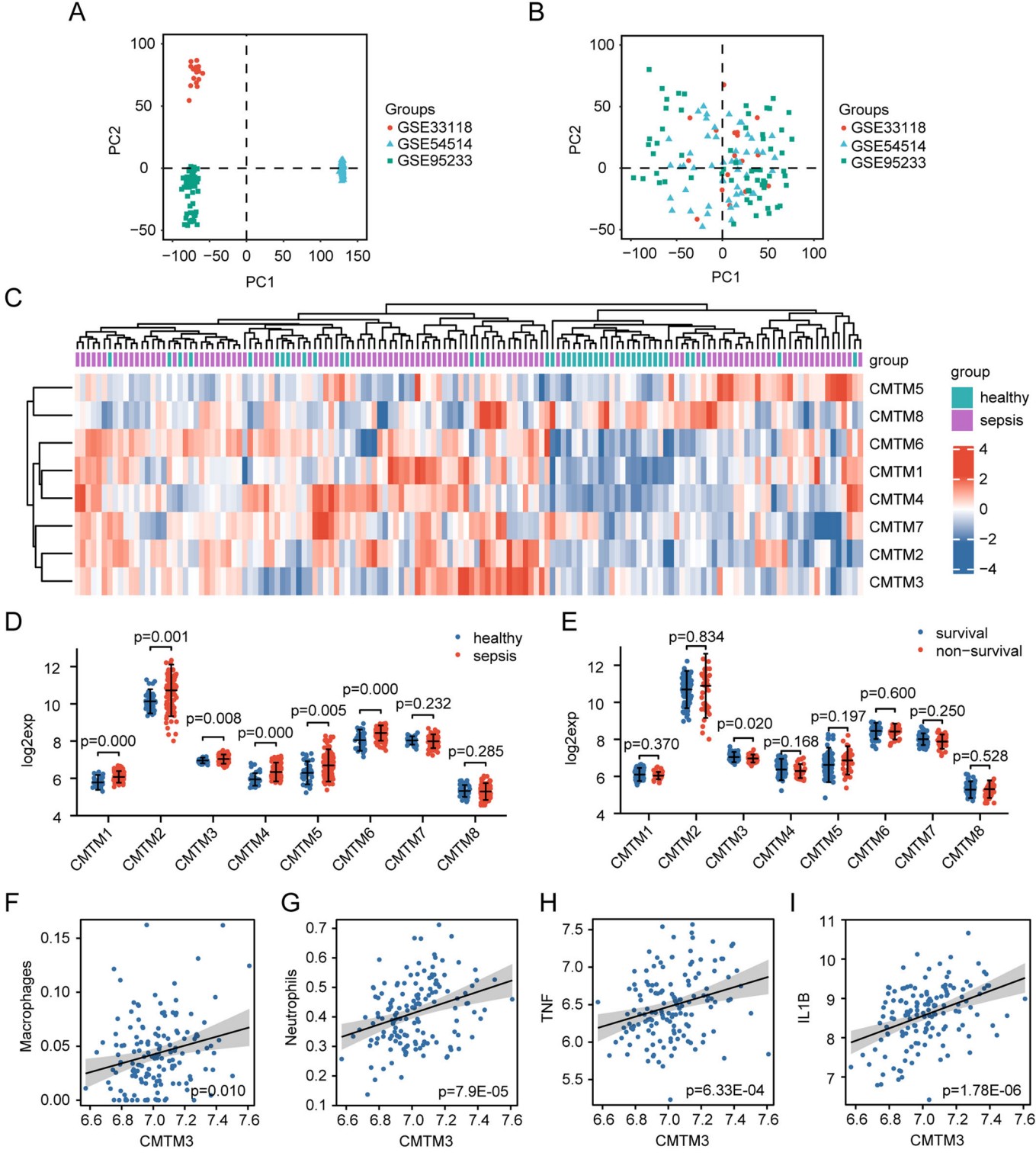

**Figure 1. Expression and correlation analysis of *CMTM* family in sepsis using public datasets.**

(A) Principal component analysis (PCA) before batch effect adjustment. (B) PCA after batch effect adjustment. (C) Expression heatmap of *CMTM1-8* in sepsis and healthy individuals. (D) Comparison of *CMTM1-8* expression differences between sepsis and healthy groups (healthy = 40, sepsis = 106). (E) Comparison of *CMTM1-8* expression differences between survival and non-survival groups (survival = 70, non-survival = 36). (F) The correlation between *CMTM3* and macrophages (Spearman analysis, r = 0.213, n = 146). (G) The correlation between *CMTM3* and neutrophils (Pearson analysis, r = 0.321, n = 146). (H) The correlation between *CMTM3* and *TNF* (Pearson analysis, r = 0.28, n = 146). (I) The correlation between *CMTM3* and *IL1B* (Spearman analysis, r = 0.383, n = 146). Data information: In (D, E), data are expressed as the median ± IQR and were analyzed by Mann–Whitney U test. Source data are available online for this figure.

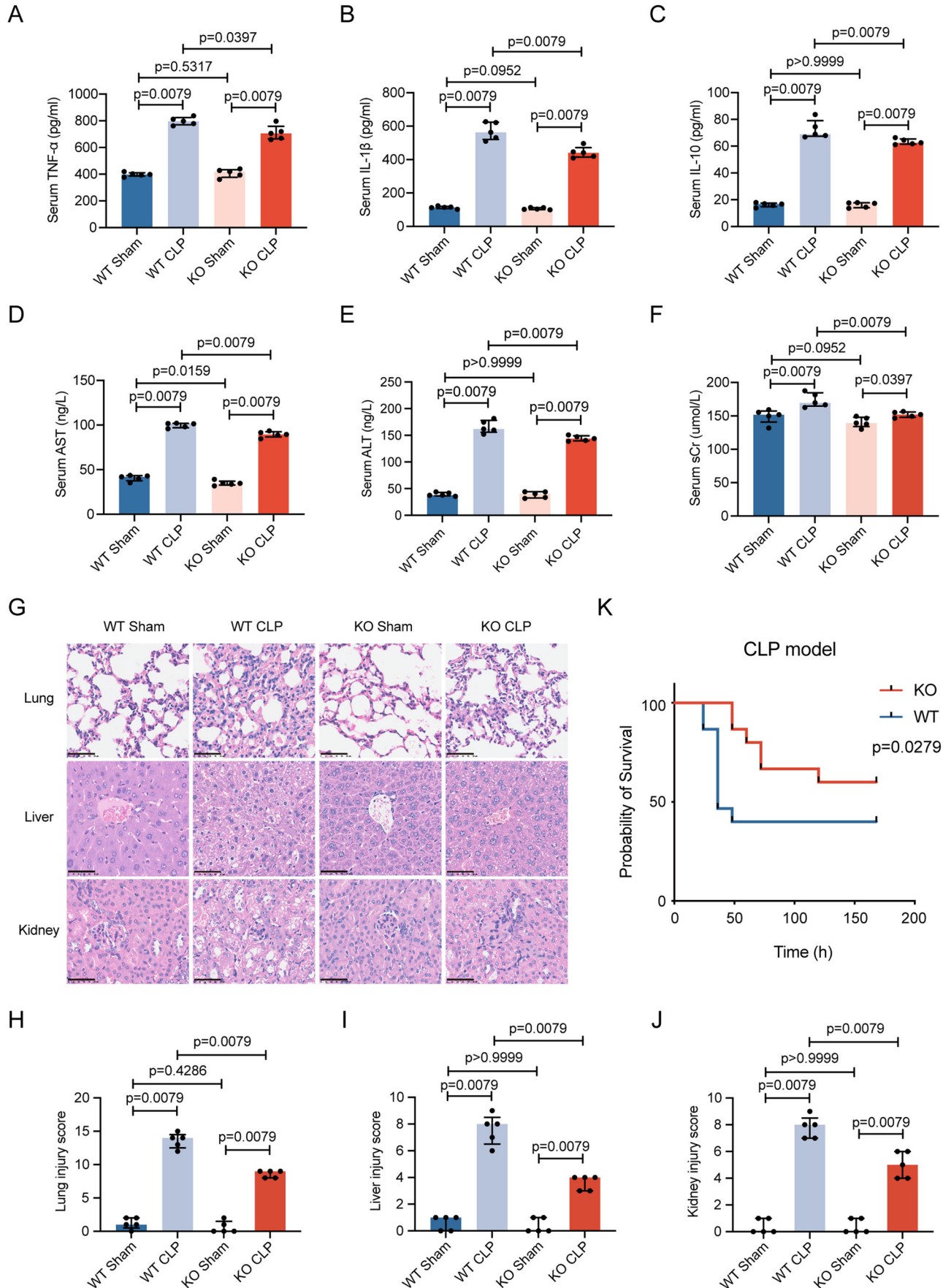

Figure 2.  The effect of *Cmtm3* KO on the severity in CLP mice.

(A–C) Serum TNF-α, IL-1β, and IL-10 were detected using ELISA 24 h after Sham or CLP (*n* = 5 mice per group). (D–F) Serum AST, ALT, and sCr were detected using ELISA 24 h after Sham or CLP (*n* = 5 mice per group). (G) HE stains of liver, lung, and kidney 24 h after Sham or CLP (scale bar: 50μm). (H–J) Injury score of liver, lung, and kidney (*n* = 5 mice per group). (K) Survival rate of WT and KO mice 7 days after CLP surgery (*n* = 15 mice per group). Data information: In (A–F, H–J), data are expressed as the median ± IQR and were analyzed by Mann–Whitney U test. Source data are available online for this figure.

of neutrophils from the bone marrow. Several chemokines, including CXCL1, CXCL2, and G-CSF, have been shown to drive the egress of neutrophils from the bone marrow in mice, leading to a rapid increase in blood neutrophil counts (Mehta and Corey, 2021). Therefore, we also investigated the expression of CXCL1, CXCL2, and G-CSF in the peripheral blood. The results showed that in the CLP model, the expression of these factors was significantly increased in the peripheral blood, while it was significantly reduced in the KO mice (Fig. 3G–I).

We also performed transcriptome sequencing on BM cells isolated from WT and KO mice (Fig. EV3A). A comparison of KO CLP versus WT CLP groups using a criterion of |log$_2$FoldChange| > 1 and *p.adj* < 0.05 for differential gene expression analysis revealed 57 upregulated genes and 209 downregulated genes (Fig. EV3B). The results of the GO enrichment analysis indicated that these down-regulated genes are associated with the cellular response to LPS, cell chemotaxis function, and cytokine production (Fig. EV3C,D).

In addition, we performed immunohistochemical staining for F4/80 to elucidate the infiltration of macrophages in the tissues. The results showed that in the CLP model, the number of macrophages in the tissues of KO mice was lower than that in WT mice (Fig. EV4A). However, in the CLP model, there was no significant difference in the number of bone marrow macrophages (Fig. EV4B,C) and peripheral blood macrophages (Fig. EV4D,E) between WT and KO group.

## CMTM3 affects the release and retention of BM neutrophils in LPS-induced endotoxemia by influencing TLR4 expression

In order to investigate the molecular mechanisms underlying CMTM3, we also conducted a protein-protein interaction (PPI) analysis on a set of 500 genes associated with CMTM3. The results of this analysis revealed a significant enrichment of the TLR4 signaling pathways, indicating its potential involvement as key pathway in mediating the effects of CMTM3 (Fig. 4A). TLR4 serves as a critical receptor for transmembrane signaling of LPS. Therefore, we further explored these pathways using a mouse model of LPS-induced endotoxemia. Consistent with our findings in the CLP model, *Cmtm3* KO resulted in a reduction of neutrophil infiltration in the liver, lung, and kidney tissues of mice with LPS-induced endotoxemia (Fig. 4B). As depicted in Fig. 4C,D, the population of peripheral blood neutrophils was significantly lower in *Cmtm3* KO mice compared to WT mice in the LPS-induced endotoxemia model. Conversely, KO mice exhibited a higher BM neutrophil count compared to WT mice in the LPS-induced endotoxemia model, as shown in Fig. 4E,F.

Furthermore, we examined the expression of TLR4 on the surface of neutrophils in both the BM (Fig. 4G,H) and peripheral blood (Fig. 4I,J). Our results demonstrate that following LPS stimulation, there was an increase in TLR4 expression on the

surface of neutrophils, with a lesser extent of increase observed in KO mice compared to WT mice.

We also conducted additional investigations in the CLP model to explore the impact of *Cmtm3* deletion on the expression of TLR4 on the surface of monocytes in both the peripheral blood (Fig. EV5A,B) and bone marrow (Fig. EV5C,D). Our findings indicate that after the induction of the CLP model, there was an observed increase in TLR4 expression on the surface of monocytes in WT mice. In contrast, the KO mice exhibited a reduced magnitude of increase in TLR4 expression on monocytes following the CLP procedure.

Then we examined whether the deletion of CMTM3 affected the expression of TLR2 on the surface of neutrophils in both peripheral blood (Fig. EV5E,F) and bone marrow (Fig. EV5G,H). Following the CLP model, there was an increase in TLR2 expression on the surface of neutrophils in WT mice, whereas the deletion of *Cmtm3* did not affect the extent of TLR2 expression increase on neutrophils. This demonstrates the specificity of the effect on TLR4 expression.

## CMTM3 regulates imbalanced neutrophil migration by modulating the expression and activation of CXCR2 through TLR4 signaling

According to the literature, TLR4 has been shown to regulate the expression of CXCR2 on the surface of neutrophils, thereby influencing their release and migration (Luo et al, 2023). Considering this, we conducted additional investigations to evaluate the expression of CXCR2 on the surface of neutrophils in both BM (Fig. 5A,B) and peripheral blood (Fig. 5C,D). Our results demonstrated that following LPS stimulation, there was an increase in CXCR2 expression on the surface of neutrophils, with a lesser extent of increase observed in KO mice compared to WT mice. Furthermore, CXCR2-specific antibodies prevented the release of BM neutrophils in both *Cmtm3* WT mice and *Cmtm3* KO mice in the LPS-induced endotoxemia model, as shown in (Fig. 5E–H). To evaluate the migration function of neutrophils, we isolated neutrophils from bone marrow (Fig. 5I) and peritoneal lavage (PL) fluid (Fig. 5J) and assessed their chemotaxis ability. The results showed that neutrophils from KO mice exhibited lower chemotaxis capability compared to WT mice with LPS pretreatment.

## Overexpression of TLR4 reverses the effects of *Cmtm3* KO on imbalanced neutrophil migration

To confirm whether CMTM3 affects the expression of CXCR2 on neutrophil membranes and abnormal migration response of neutrophils in septic mice by regulating TLR4, we overexpressed TLR4 in *Cmtm3* KO mice. We found that compared to the empty vector control group, the overexpression of TLR4 successfully

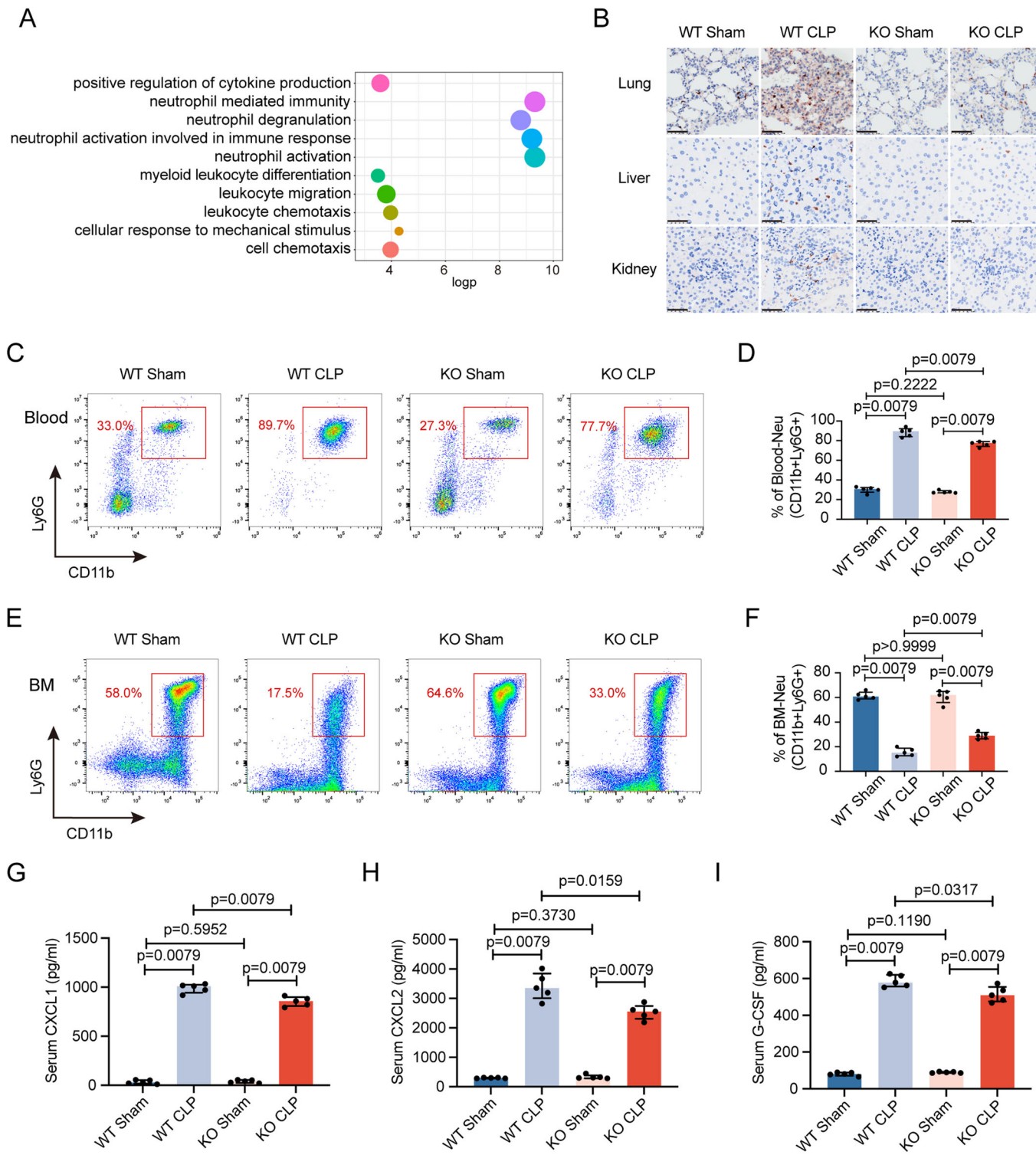

**Figure 3. The effect of *Cmtm3* KO on the release and retention of BM neutrophils in CLP mice.**

(A) GO-BP enrichment analysis of 500 genes highly correlated with *CMTM3* expression pattern. (B) IHC staining of Ly6G of liver, lung, and kidney 24 h after Sham or CLP (scale bar: 50 μm). (C, D) Blood neutrophil populations in WT and KO mice 24 h after Sham or CLP ($n = 5$ mice per group). (E, F) BM neutrophil populations in WT and KO mice 24 h after Sham or CLP ($n = 5$ mice per group). (G–I) Serum CXCL1, CXCL2, and G-CSF were detected using ELISA 24 h after Sham or CLP ($n = 5$ mice per group). Data information: In (A), data were analyzed by Hypergeometric test. In (D, F–I), data are expressed as the median ± IQR and were analyzed by Mann–Whitney U test. Source data are available online for this figure.

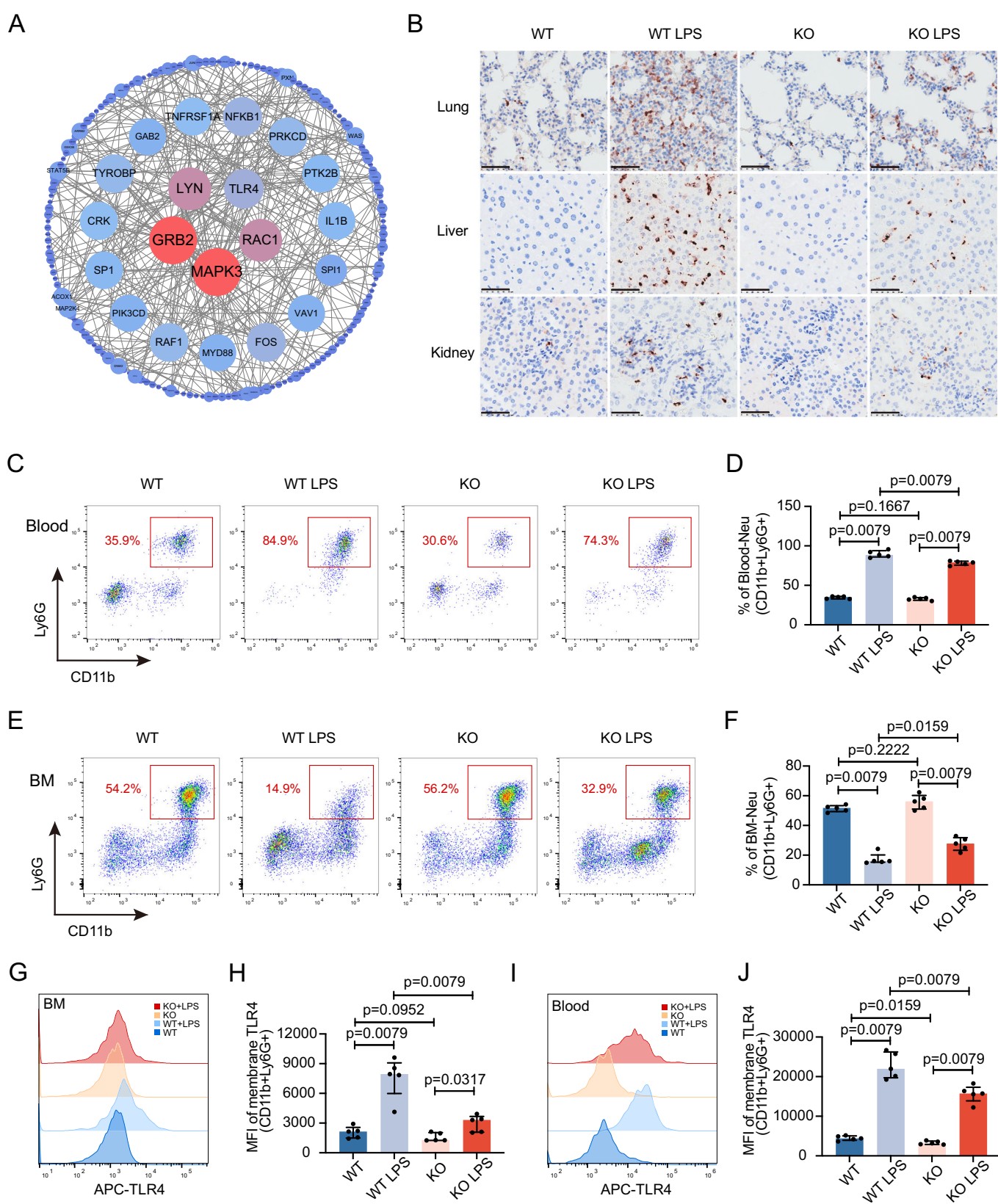

**Figure 4. The effect of *Cmtm3* KO on the neutrophil migration imbalance and TLR4 expression in LPS-induced endotoxemia.**

(A) PPI analysis of 500 genes highly correlated with *CMTM3* expression pattern. (B) IHC staining of Ly6G of liver, lung, and kidney 24 h after PBS or LPS injection (scale bar: 50 μm). (C, D) Blood neutrophil populations in WT and KO mice 24 h after PBS or LPS treatment ($n = 5$ mice per group). (E, F) BM neutrophil populations in WT and KO mice 24 h after PBS or LPS treatment ($n = 5$ mice per group). (G, H) TLR4 membrane expression in BM neutrophils in WT and KO mice 24 h after PBS or LPS treatment ($n = 5$ mice per group). (I, J) TLR4 membrane expression in blood neutrophils in WT and KO mice 24 h after PBS or LPS treatment ($n = 5$ mice per group). Data information: In (D, F, H, J), data are expressed as the median ± IQR and were analyzed by Mann–Whitney U test. Source data are available online for this figure.

upregulated the expression of TLR4 on the surface of BM (Fig. 6A,B) and peripheral blood (Fig. 6C,D) neutrophils in *Cmtm3* KO mice. At the same time, the overexpression of TLR4 also successfully upregulated the expression of CXCR2 on the surface of BM (Fig. 6E,F) and peripheral blood (Fig. 6G,H) neutrophils. We further detected the distribution proportion of neutrophils, and the results confirmed that the overexpression of TLR4 successfully rescued the retention effect of *Cmtm3* knockout on BM neutrophils. Obviously, compared with the empty vector control group, the overexpression of TLR4 reduced the population of BM neutrophils in KO mice (Fig. 6I,J), while significantly increasing the population of peripheral blood neutrophils (Fig. 6K,L).

## Overexpression of TLR4 reverses the protective effect of *Cmtm3* KO in LPS-induced endotoxemia

Next, we validated the improvement of disease severity by *Cmtm3* knockout through the TLR4 pathway in an LPS-induced endotoxemia model. As shown in Fig. 7A–C, *Cmtm3* KO alleviated the release of TNF-α, IL-1β, and IL-10 in the peripheral blood of LPS-induced endotoxemia mice, while TLR4 overexpression reversed this alleviating effect. In addition, in LPS-induced endotoxemia mice, the expression levels of AST, ALT, and sCr in the peripheral blood were lower in KO mice compared to WT mice, while the overexpression of TLR4 resulted in corresponding increases in the expression levels of these organ injury markers (Fig. 7D–F). To evaluate tissue histopathological damage, we examined liver, lung, and kidney tissues again using HE stains. The results showed that *Cmtm3* KO reduced the pathological injury in the liver, lungs, and kidneys of LPS-induced endotoxemia mice, while TLR4 overexpression reversed this alleviating effect (Fig. 7G). Figure 7H–J provide the quantitative scoring of the injury severity in these organs, respectively. Survival analysis indicated that *Cmtm3* KO significantly improved the survival rate of LPS-induced endotoxemia mice, while TLR4 overexpression increased the mortality rate of *Cmtm3* KO mice (Fig. 7K).

## Overexpression of TLR4 reverses the protective effect of *Cmtm3* KO in the CLP model

Similarly, we further validated the protective effect of *Cmtm3* knockout through the TLR4 pathway in a CLP-induced sepsis model. As shown in Fig. 8A–C, *Cmtm3* KO reduced the release of TNF-α, IL-1β, and IL-10 in the peripheral blood of CLP-induced sepsis mice, while TLR4 overexpression reversed this protective effect. In addition, in CLP-induced sepsis mice, the expression levels of AST, ALT, and sCr in the peripheral blood were lower in KO mice compared to WT mice, whereas the overexpression of TLR4 resulted in corresponding increases in these organ injury markers (Fig. 8D–F). Histopathological analysis of liver, lung, and

kidney tissues using HE stains revealed that *Cmtm3* KO reduced the pathological injury in these organs in CLP-induced sepsis mice, while TLR4 overexpression reversed this effect (Fig. 8G). Figure 8H–J provides the quantitative scoring of the injury severity in these organs, respectively. Survival analysis indicated that *Cmtm3* KO significantly improved the survival rate of CLP-induced sepsis mice, whereas TLR4 overexpression increased the mortality rate of *Cmtm3* KO mice (Fig. 8K).

## Discussion

The appropriate activation and migration of neutrophils play a critical role in infection control. Modulating the dysregulation of neutrophil migration holds potential for the development of promising therapeutic strategies for the treatment of sepsis. In this investigation, we identified CMTM3 as a potential contributor to the pathogenesis of sepsis. Our findings demonstrate that the deletion of *Cmtm3* leads to an improvement in survival rates, reduction in inflammatory response, and mitigation of organ damage in both CLP and LPS-induced mice models. Mechanistically, CMTM3 regulates the equilibrium of circulating and tissue neutrophils by modulating the expression of CXCR2 on neutrophil membranes in a TLR4-dependent manner. Notably, the ablation of *Cmtm3* diminishes neutrophil infiltration in liver, lung, and kidney tissues of septic mice, thereby alleviating the ensuing inflammatory response. Moreover, *Cmtm3* deletion was shown to decrease neutrophil mobilization from the bone marrow, likely due to the downregulation of cytokine release, highlighting CMTM3 as a potential target for anti-inflammatory therapy in sepsis.

The CMTM family exhibits widespread expression within the immune system, with dysregulated expression linked to various tumors and autoimmune diseases (Duan et al, 2020; Ge et al, 2021; Wu et al, 2020). However, the expression and function of CMTM family members in sepsis have not been reported previously. Our comprehensive bioinformatics analysis revealed that *CMTM1-6* are upregulated in sepsis, with CMTM3 showing differential expression between survival and non-survival groups, suggesting a distinct role for CMTM3 in sepsis progression. Intriguingly, our study found higher CMTM3 expression in sepsis survivors, which may reflect an adaptive immune response that mitigates the severity of sepsis. This observation, although seemingly counterintuitive given the protective effect of *Cmtm3* knockout in septic mice, underscores the complexity of sepsis pathophysiology, where gene expression does not linearly predict outcomes. It is plausible that higher CMTM3 expression in survivors represents an immune adaptation that, while beneficial in some contexts, could exacerbate disease severity in others. Further research is needed to fully elucidate the role of CMTM3 in sepsis.

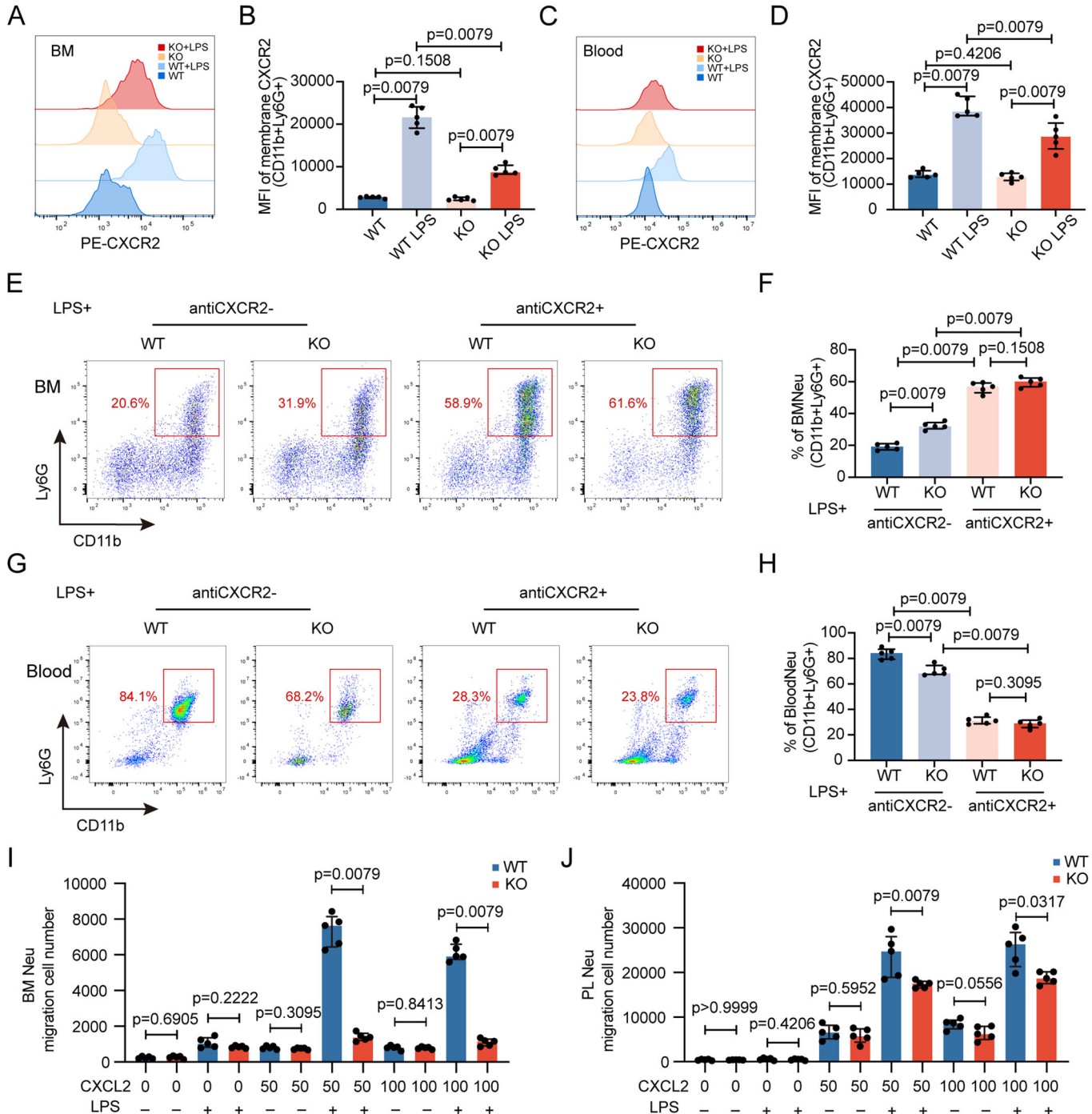

**Figure 5. The effect of *Cmtm3* KO on the expression and activation of neutrophil CXCR2 in LPS-induced endotoxemia.**

(A, B) CXCR2 membrane expression in BM neutrophils in WT and KO mice 24 h after PBS or LPS treatment ($n = 5$ mice per group). (C, D) CXCR2 membrane expression in blood neutrophils in WT and KO mice 24 h after PBS or LPS treatment ($n = 5$ mice per group). (E, F) BM neutrophil populations in WT and KO mice 24 h after LPS injection with or without antiCXCR2 pretreatment ($n = 5$ mice per group). (G, H) Blood neutrophil populations in WT and KO mice 24 h after LPS injection with or without antiCXCR2 pretreatment ($n = 5$ mice per group). (I) The chemotactic activity of BM neutrophils from WT and KO mice with or without LPS pretreatment ($n = 5$ mice per group). (J) The chemotactic activity of PL neutrophils from WT and KO mice with or without LPS pretreatment ($n = 5$ mice per group). Data information: In (B, D, F, H–J), data are expressed as the median ± IQR and were analyzed by Mann–Whitney U test. Source data are available online for this figure.

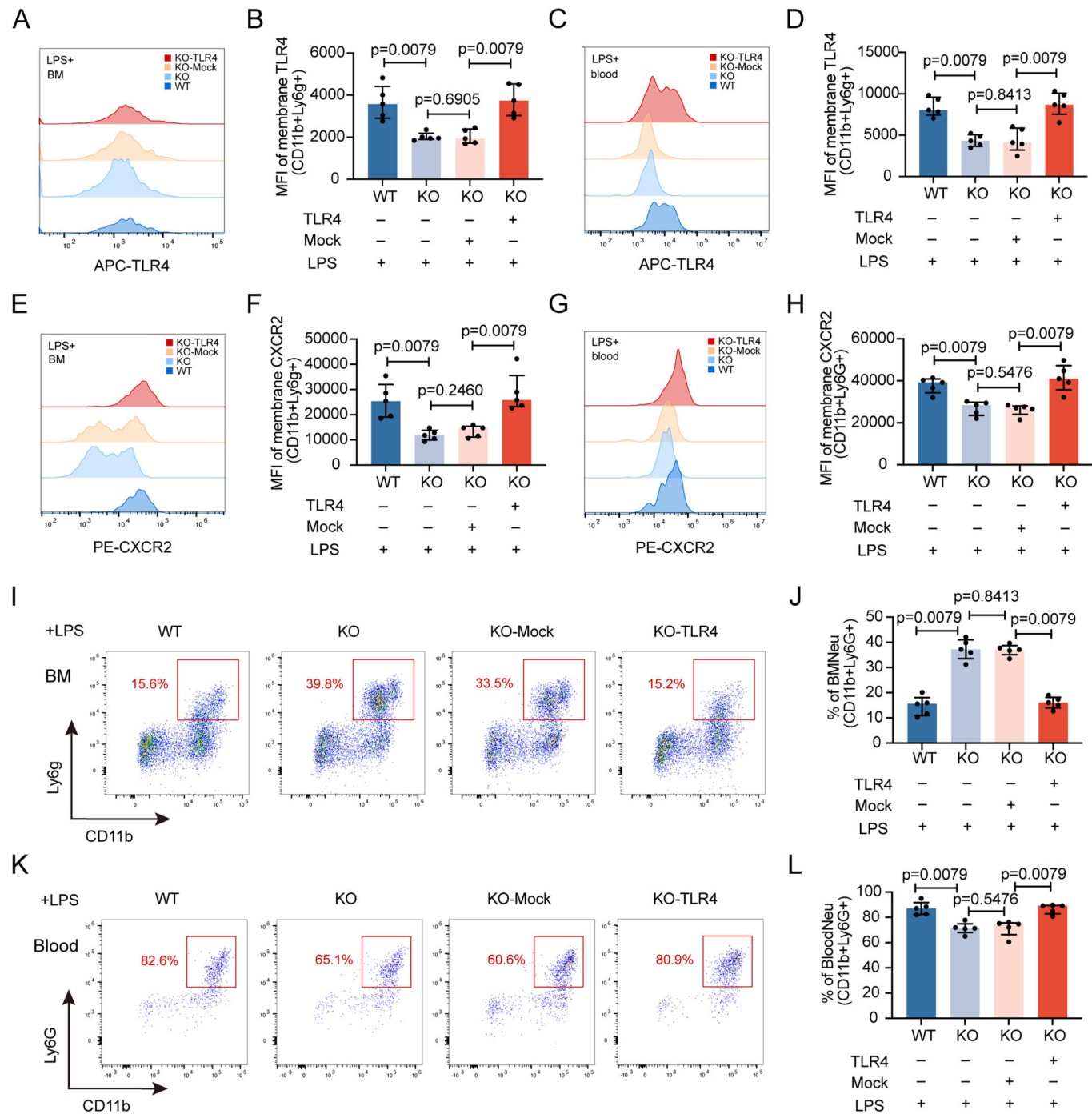

**Figure 6. The effect of overexpressing TLR4 in *Cmtm3* KO mice on neutrophil migration imbalance in LPS-induced endotoxemia.**

(A, B) TLR4 membrane expression in BM neutrophils with or without TLR4 overexpression ($n = 5$ mice per group). (C, D) TLR4 membrane expression in blood neutrophils with or without TLR4 overexpression ($n = 5$ mice per group). (E, F) CXCR2 membrane expression in BM neutrophils with or without TLR4 overexpression ($n = 5$ mice per group). (G, H) CXCR2 membrane expression in blood neutrophils with or without TLR4 overexpression ($n = 5$ mice per group). (I, J) BM neutrophil populations with or without TLR4 overexpression ($n = 5$ mice per group). (K, L) Blood neutrophil populations with or without TLR4 overexpression ($n = 5$ mice per group). Data information: In (B, D, F, H, J, L), data are expressed as the median ± IQR and were analyzed by Mann–Whitney U test. Source data are available online for this figure.

In the context of sepsis, genes exert their influence through co-expression gene networks that possess similar biological functions (van Dam et al, 2018). Our analysis of genes co-expressed with CMTM3 revealed a significant association with neutrophil chemotaxis. Using a *Cmtm3* knockout mice model, we observed that the deletion of *Cmtm3* ameliorates dysregulated neutrophil migration and reduces neutrophil infiltration in vital organs during sepsis. These findings emphasize the role of neutrophils as principal

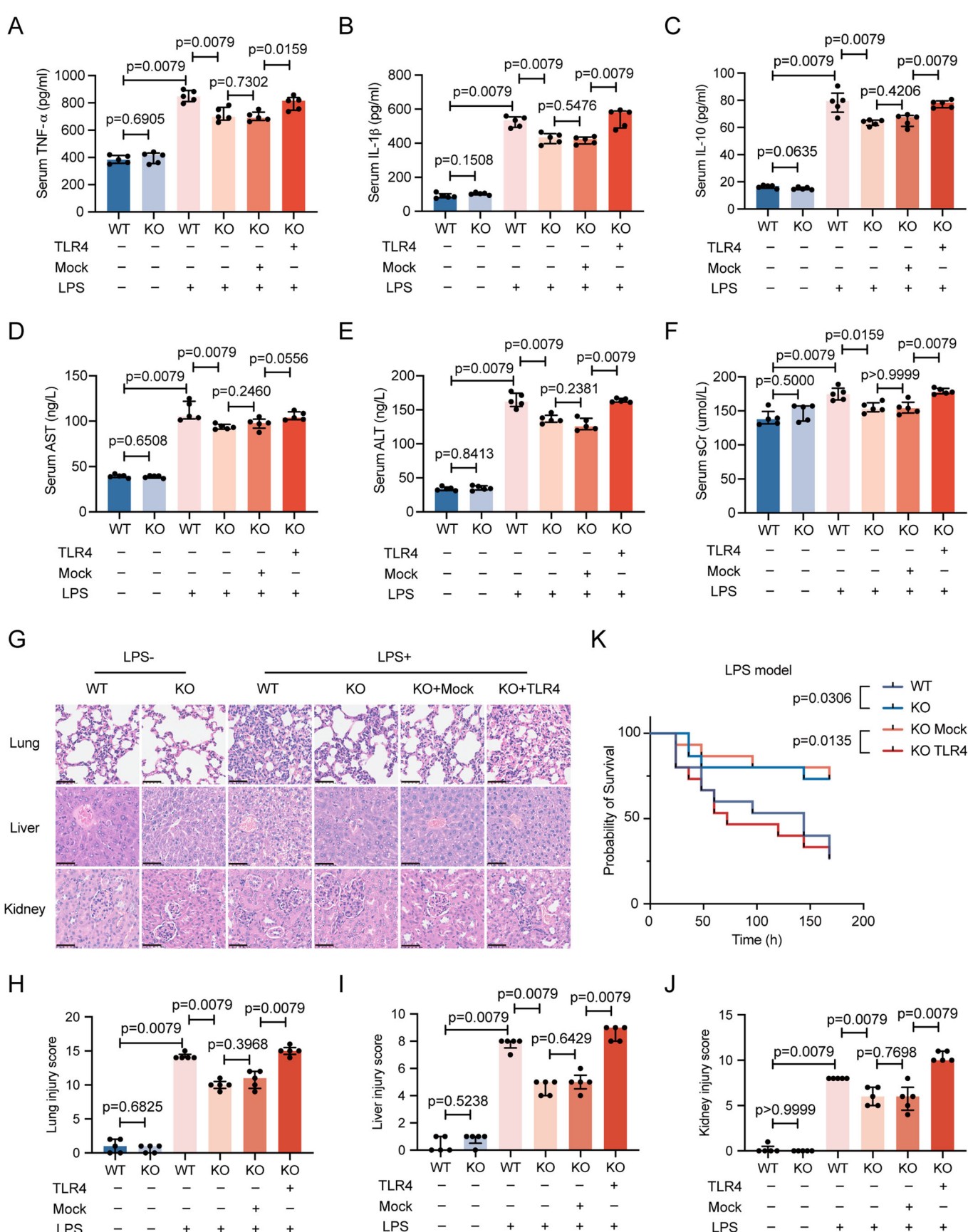

◄ **Figure 7. Overexpression of TLR4 reverses the protective effect of *Cmtm3* KO in LPS-induced endotoxemia.**

(A–C) Serum TNF-α, IL-1β, and IL-10 were detected 24 h after PBS or LPS treatment (*n* = 5 mice per group). (D–F) Serum AST, ALT, and sCr were detected 24 h after PBS or LPS treatment (*n* = 5 mice per group). (G) HE stains of liver, lung, and kidney 24 h after PBS or LPS treatment (scale bar: 50μm). (H–J) Injury score of liver, lung, and kidney (*n* = 5 mice per group). (K) Survival rate of WT and KO mice 7 days after LPS treatment (*n* = 15 mice per group). Data information: In (A–F, H–J), data are expressed as the median ± IQR and were analyzed by Mann–Whitney U test. Source data are available online for this figure.

cellular targets through which CMTM3 exerts its effects in sepsis. However, it is important to note that our systemic knockout approach may not fully capture the cell-specific effects of CMTM3. Future studies utilizing conditional knockout models are necessary to dissect the cell-type specific contributions of CMTM3, which will provide a clearer understanding of its role in sepsis.

The literature indicates that TLR4 plays a crucial role in neutrophil migration during sepsis, influencing neutrophil movement through the regulation of CXCR2 expression (Alves-Filho et al, 2006; Fan and Malik, 2003). The overactivation of the TLR4 signaling pathway is a common feature in sepsis, and controlling its expression is considered key to regulating neutrophil activity (Wittebole et al, 2010). Our study suggests that CMTM3 may be involved in the regulation of TLR4 expression, thereby linking it to innate immune responses. Given that the CMTM3-TLR4 axis may have a conserved survival benefit in biology (Branger et al, 2004), further studies should investigate the impact of CMTM3 on neutrophil egress and pathogen clearance in a CLP model without the use of antibiotics, to gain a more comprehensive understanding of the adaptive functions of this pathway and its role in sepsis pathophysiology. In addition, future research should focus on elucidating the precise molecular mechanisms by which CMTM3 regulates TLR4 expression, to clarify its dual role in sepsis.

CXCR2 is a critical chemokine receptor expressed on neutrophils, plays a key role in their maturation and mobilization from the bone marrow (Eash et al, 2010). Our research has shown that CMTM3 regulates CXCR2 expression through the TLR4 pathway. Neutrophils from *Cmtm3* knockout mice exhibited lower CXCR2 expression, leading to increased retention of neutrophils in the bone marrow. This aligns with our hypothesis that CMTM3 influences neutrophil chemotaxis. However, neutrophils are a heterogeneous population, and their maturation and differentiation are influenced by multiple factors (Furze and Rankin, 2008; Qi et al, 2021). Thus, our study does not definitively determine whether CMTM3 affects neutrophil maturation, necessitating further research in this area.

There are also limitations to this study. For example, the exclusive use of male mice, chosen for hormone stability and suitable body size, which might mask sex-specific responses to CMTM3 deletion. Besides, while our findings suggest that CMTM3 regulates TLR4 expression and neutrophil migration, the precise molecular mechanisms remain unclear. Further research should address these gaps to fully understand the role of CMTM3 in sepsis.

Taken together, our study proposes that CMTM3 regulates the expression of CXCR2 on neutrophils through the TLR4 pathway, resulting in a reduced migration of neutrophils from the bone marrow to the bloodstream and thereby decreasing their recruitment to vital organs. Targeting CMTM3 thus could improve organ damage and survival rates in septic mice, thus potentially serving as a novel therapeutic target for the treatment of sepsis.

# Methods

### Reagents and tools table

| Reagent/Resource | Reference or Source | Identifier or Catalog Number |
|---|---|---|
| **Experimental Models** | | |
| Cmtm3$^{-/-}$ C57/BL6 mice | Professor Han Wenling's laboratory at Peking University Health Science Center | N/A |
| Cmtm3$^{+/+}$ C57/BL6 mice | Professor Han Wenling's laboratory at Peking University Health Science Center | N/A |
| **Recombinant DNA** | | |
| LV-Tlr4(24379-1) | Genechem Co. | N/A |
| CON335 | Genechem Co. | LVCON335 |
| **Antibodies** | | |
| APC/Cyanine7 anti-mouse CD45 | Biolegend | 103116 |
| FITC anti-mouse/human CD11b | Biolegend | 101205 |
| Brilliant Violet 785 anti-mouse Ly-6G | Biolegend | 127645 |
| PE anti-mouse CD182 (CXCR2) | Biolegend | 149304 |
| PE/Cyanine7 anti-mouse F4/80 | Biolegend | 123114 |
| APC anti-mouse CD284 (TLR4) | Biolegend | 145406 |
| 7-AAD Viability Staining Solution | Biolegend | 420403 |
| Anti-Ly6G Rabbit pAb | Servicebio | GB11229 |
| F4/80 (D2S9R) XP® Rabbit mAb | CST | 70076 |
| Goat Anti-Rabbit IgG H&L (HRP) | Abcam | ab205718 |
| Mouse CXCR2/IL-8RB Antibody | R&D Systems | MAB2164 |
| **Chemicals, Enzymes and other reagents** | | |
| Mouse TNF-α ELISA Kit | MEIMIAN | MM-0132 |
| Mouse IL-1β ELISA KIT | MEIMIAN | MM-0905 |
| Mouse IL-10 ELISA KIT | MEIMIAN | MM-0176 |
| Mouse AST ELISA KIT | MEIMIAN | MM-44115 |
| Mouse ALT ELISA KIT | MEIMIAN | MM-0260 |
| Mouse sCr ELISA KIT | MEIMIAN | MM-44455 |
| Mouse CXCL1 ELISA KIT | MEIMIAN | MM-43835 |

| Reagent/Resource | Reference or Source | Identifier or Catalog Number |
|---|---|---|
| Mouse CXCL2 ELISA KIT | MEIMIAN | MM-1012 |
| G-CSF ELISA KIT | MEIMIAN | MM-0186 |
| Recombinant Mouse CXCL2 Protein | R&D Systems | 452-M2 |
| Bovine Serum Albumin | Solarbio | A8010 |
| Sodium thioglycolate | Sigma-Aldrich | T0632 |
| Lipopolysaccharide | Solarbio | L8880 |
| 1× RBC Lysis Buffer | Invitrogen™ | 00-4333-57 |
| TRIzol® Reagent | Invitrogen™ | 15596026 |
| RevertAid First Strand cDNA Synthesis Kit | Thermo Scientific | K1622 |
| SYBR® Green real-time PCR master mix | TOYOBO | QPK-201 |
| Mouse Neutrophil Isolation Kit | Solarbio | P8550 |
| 40 μm cell filter | Solarbio | F8200 |
| 24 mm Transwell with 3 μm pore polyester membrane insert | Corning® | 3452 |
| Software | | |
| RStudio v 3.6.1 | Posit | https://posit.co/products/open-source/rstudio/ |
| FlowJo v 10.8.1 | TreeStar | https://www.flowjo.com/ |
| GraphPad Prism v 9 | GraphPad Software | https://imaris.oxinst.com/packages |

## Public data acquisition and processing

A comprehensive search was conducted in the GEO database (https://www.ncbi.nlm.nih.gov/geo/) using the keyword "sepsis" to extract datasets pertaining to our study (Barrett et al, 2005). Subsequently, an extensive screening process was employed to identify datasets that featured survival information of sepsis patients as well as contained expression data of *CMTMs*. Following the acquisition of gene expression values for samples, the integration of these values was performed using the combat function from the SVA R package. This robust integration method effectively eliminated batch effects in the datasets, ensuring reliable and unbiased results. For the analysis of immune cell infiltration status, the CIBERSORT algorithm was employed. For gene ontology (GO) enrichment analysis, the ClusterProfiler R package was utilized, followed by visualization using ggplot2 R package (Ashburner et al, 2000). The STRING (v.11.5) online database at https://cn.string-db.org was employed to predict relationships between genes and construct a protein-protein interaction (PPI) network (Szklarczyk et al, 2017). Further processing and visualization of the network were performed using Cytoscape (v.3.9.1).

## Mice and materials

Ten-week-old parental *Cmtm3* gene knockout mice and their wild-type counterparts, sharing the same genetic background, were sourced from Professor Han Wenling's laboratory at the Department of Immunology, Peking University Health Science Center. These mice were originally of the same genetic background and were subjected to a controlled breeding program to ensure genetic consistency. All mice were housed in the Animal Center of Peking University People's Hospital under specific pathogen-free (SPF) conditions. The environmental conditions were strictly controlled with a temperature range of 22 to 25 °C, humidity range of 50% to 60%, and a 12-h light-dark cycle. Mice had free access to food and water. Mice aged 8 to 10 weeks, with body weights ranging from 18 to 20 g, were utilized for the experiments. Every effort was made to ensure age and weight matching among the mice in the experimental groups. All Institutional and National Guidelines for the care and use of animals were followed. The experimental protocol was reviewed and approved by the Ethics Committee for Animal Experiments of Peking University People's Hospital (Approval No. 2022PHE058).

## Cecal ligation and puncture (CLP)-induced sepsis model

The mouse was placed in a supine position and immobilized on the surgical table. Following induction of general anesthesia and administration of analgesia, the abdominal cavity was carefully opened to expose the cecum. At the midpoint of the cecum, a ligature was meticulously applied, and a single puncture was created using a 21-gauge needle. Subsequently, a small quantity of feces was extruded through the puncture site, and the cecum was repositioned within the abdominal cavity. The abdominal cavity was then meticulously closed utilizing a single knot suture technique. Post-surgery, the mouse was carefully transferred to a recovery cage and allowed to regain full consciousness. Two hours after the surgical procedure, a broad-spectrum antibiotic (Ampicillin, 50 mg/kg/12 h) was administered via intraperitoneal injection (Drechsler and Osuchowski, 2021; Rittirsch et al, 2009).

## Lipopolysaccharide (LPS)-induced endotoxemia model

The endotoxemia model was constructed by intraperitoneally injecting *Cmtm3* WT and KO mice of identical age with 10 mg/kg LPS. The control group received an equivalent volume of phosphate-buffered saline (PBS) via intraperitoneal injection. Subsequent to drug administration, the mice were placed in individual cages and carefully observed for changes in mental alertness and physical activity.

## Isolation and preparation of mouse neutrophils

In order to isolate bone marrow cells, bone marrow was extracted from the femurs and tibias of mice, followed by filtration through a 40 μm cell filter to obtain a single-cell suspension. For peripheral blood cell isolation, the retro-orbital bleeding method was employed to collect whole blood from mice into Ethylene Diamine Tetraacetic Acid (EDTA) anticoagulant-treated collection tubes. After the removal of red blood cells using a red blood cell lysis buffer, a single-cell suspension was prepared. As for peritoneal cell collection, mice were intraperitoneally injected with 1 ml of sterile 3% thioglycolate solution. After a period of 2.5 h, the peritoneal cavity was lavaged with 5 ml of

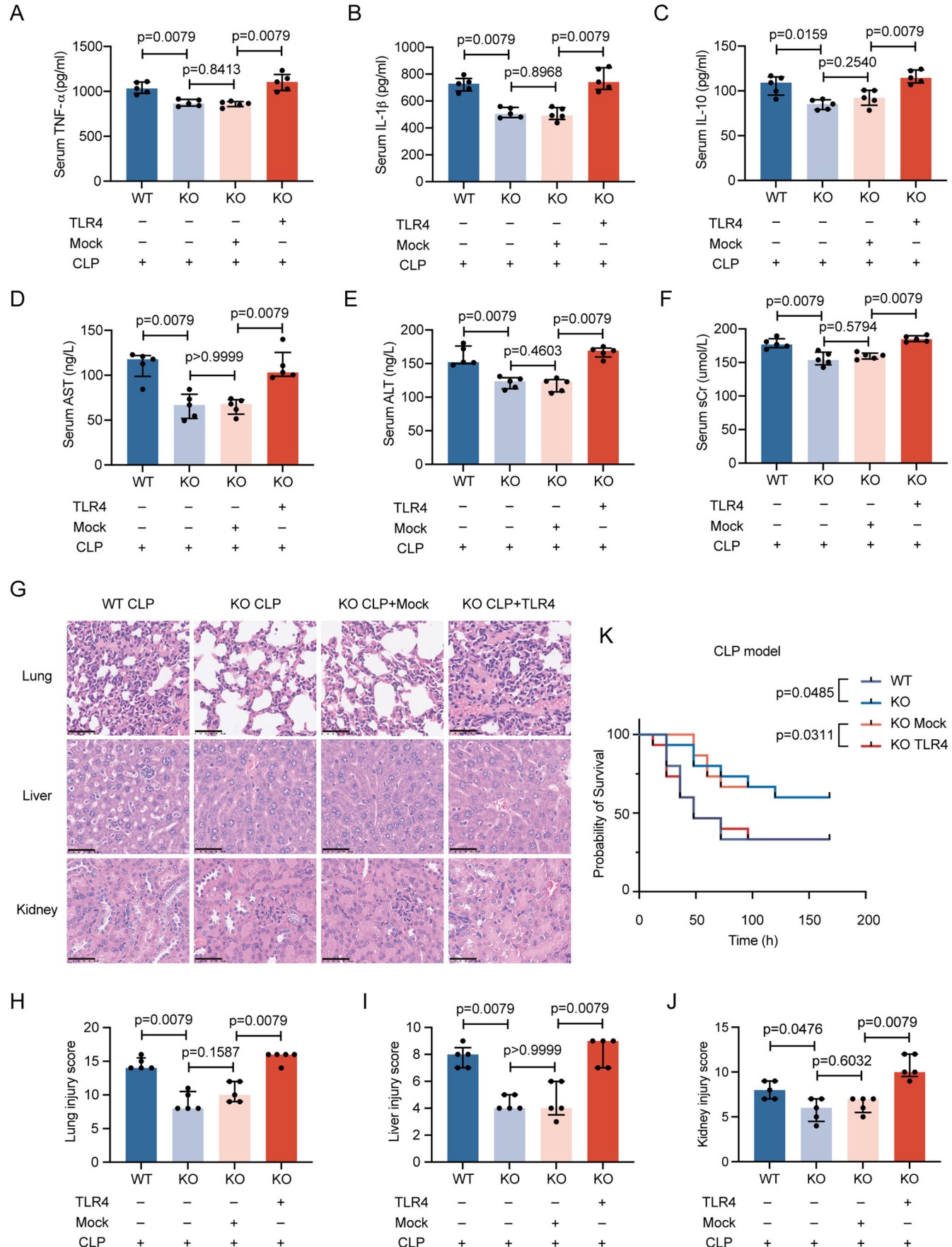

**Figure 8. Overexpression of TLR4 reverses the protective effect of *Cmtm3* KO in the CLP model.**

(A–C) Serum TNF-α, IL-1β, and IL-10 were detected 24 h after Sham or CLP (*n* = 5 mice per group). (D–F) Serum AST, ALT, and sCr were detected 24 h after Sham or CLP (*n* = 5 mice per group). (G) HE stains of liver, lung, and kidney 24 h after Sham or CLP (scale bar: 50 μm). (H–J) Injury score of liver, lung, and kidney (*n* = 5 mice per group). (K) Survival rate of WT and KO mice 7 days after CLP (*n* = 15 mice per group). Data information: In (A–F, H–J), data are expressed as the median ± IQR and were analyzed by Mann–Whitney U test. Source data are available online for this figure.

pre-chilled PBS for peritoneal flushing to collect the elicited cells. Following cell suspension acquisition, neutrophil isolation was performed using a neutrophil isolation reagent kit. In brief, neutrophils were enriched using a non-continuous density gradient composed of two different density solutions.

## Transcriptome sequencing

Total ribonucleic acid (RNA) was isolated from BM cells using Trizol reagent. Subsequently, mRNA sample libraries were constructed and subjected to high-throughput sequencing by Shanghai Sequen Gene-tech Co., LTD (China). In brief, after the quantification and qualification of total RNA samples, mRNA was purified using poly-T oligo-attached magnetic beads. The enriched mRNA was then fragmented into short fragments and the fragmented mRNA was used as a template for library construction. Once the libraries passed quality control, they were pooled based on their effective concentrations and the desired amount of data for sequencing on the Illumina Hiseq platform.

## Flow cytometry analysis

Following resuspension of cells in cell wash buffer containing 2% Bovine Serum Albumin (BSA) in PBS, the cells were carefully stained with specific target antibodies for a duration of 30 min. Samples devoid of any antibodies were used as blank controls, while samples with isotype antibodies served as isotype controls. In addition, samples with a single target antibody were designated as compensation controls. Once the cells were appropriately stained, they were washed three times for optimal cleanliness and then resuspended for acquisition using a Beckman Coulter CytoFLEX flow cytometer. Subsequently, the data obtained was analyzed using FlowJo software (v.10.8.1). The staining and gating approach employed entailed identification of neutrophils within the CD45+ cell population as Ly6G + CD11b+ cells, and monocytes/macrophages as F4/80 + CD11b+ cells.

## Construction of lentiviral vector for TLR4 overexpression (LV-TLR4)

The lentiviral vector overexpressing TLR4 (NM_021297) was constructed by Shanghai Genechem Co., Ltd. To elaborate, The GV492 plasmid (Ubi-MCS-3FLAG-CBh-gcGFP-IRES-puromycin), which encodes TLR4 overexpression at the BamHI/AgeI sites, was transfected into 293T cells, yielding the lentiviral particles. The lentiviral particles were meticulously collected, filtered through a 0.45-mm MCE membrane, and subsequently employed to infect bone marrow cells in the femur. For optimal bone marrow cell infection in mouse femurs, a 40 μl mixture of LV-TLR4 ($10^8$ IFU) or the lentiviral empty vector (Mock) was injected directly into the femur using a 30-gauge insulin needle through the kneecap. Flow

cytometry was employed to assess the efficiency of TLR4 overexpression.

## Blockade of CXCR2 in vivo

A total volume of 100 μl sterile PBS containing 20 μg mouse CXCR2 antibody was injected into the peritoneal cavity. Two hours later, LPS was injected intraperitoneally. Cells from bone marrow and blood were collected for flow cytometry analysis 24 h after LPS injection.

## Transwell assay

Chemotactic activity was assessed using Corning Costar Transwell® cell culture inserts with a pore size of 3 μm. In brief, $3 \times 10^5$ neutrophils were either untreated or pre-treated with 100 ng/ml LPS for 1 h, and the suspension was plated onto the transwell inserts. The lower chamber of the Transwell device contained culture medium with or without different concentration of recombinant mouse C-X-C motif chemokine ligand (CXCL) 2. The Transwell system was then placed in a 37 °C, 5% $CO_2$ incubator, and after 1 h, the number of migrated cells in the lower chamber was assessed using flow cytometry analysis.

## Enzyme-linked immunosorbent assay (ELISA)

Mouse serum was collected at designated time points. Cytokine concentrations were measured using ELISA kits specific for mouse Interleukin (IL) 1β, Tumor necrosis factor (TNF)α, and IL10. Concentrations of organ injury markers including mouse aspartate aransaminase (AST), mouse alanine aminotransferase (ALT), and mouse serum creatinine (sCr) were quantified using ELISA kits. The same method was employed to detect chemokines CXCL1, CXCL2, and Granulocyte colony stimulating factor (G-CSF). All ELISA assays were performed according to the manufacturer's instructions.

## Histology and immunohistochemistry (IHC)

After the completion of the experiment, euthanasia was performed on the mice. Liver, lung, and kidney tissues were harvested and washed with PBS solution to remove blood. Subsequently, a portion of each tissue was immersed in 4% paraformaldehyde fixative solution and kept at 4 °C overnight. On the following day, tissue samples were transferred to 70–100% ethanol for dehydration and then embedded in paraffin. Tissue sections (5 μm apart) were dewaxed and stained with hematoxylin and eosin (H&E) according to standard procedures. For IHC, slides were deparaffinized, rehydrated, and subjected to antigen retrieval in a pressure cooker using antigen retrieval solution (pH 9.0). Endogenous peroxidase activity was quenched for 20 min using 3% hydrogen peroxide in PBS. The sections were then blocked with 10% goat serum. Slides

were incubated with anti-Lyg6 antibody (diluted 1:1000) or anti-Lyg6 antibody (diluted 1:800) at 4 °C overnight. On the following day, they were incubated with the second antibody. Finally, 3,3′-diaminodbenzidine (DAB) substrate was used for staining, followed by counterstaining with hematoxylin.

## Quantification and statistical analysis

Data processing and statistical analyses in this study were conducted using R software (version 3.6.1), and GraphPad Prism (version 9). Data that followed a normal distribution were analyzed with two-tailed t-tests to calculate *p*-values. For data that did not adhere to normal distribution or when the distribution status was unknown, the Mann–Whitney U test was employed to evaluate differences between two independent groups. Summary bar graphs were designed to visually represent individual data points as dots. The horizontal lines within each bar depicted the mean value for normally distributed data and the median value for non-normally distributed data analyzed with the Mann–Whitney U test. Error bars indicated the standard error of the mean (SEM) for normally distributed data, and the interquartile range (IQR) or standard deviation (SD), as determined by the data distribution, for non-normally distributed data. Correlation analyses were performed using Pearson's correlation analysis for datasets that met the assumptions of normality. For datasets that either did not meet these assumptions or were ordinal in nature, Spearman's rank correlation analysis was applied. In survival analysis, the Gehan-Breslow-Wilcoxon test was chosen. Statistical significance was denoted by exact *p*-values.

## Data availability

The transcriptome sequencing data used in this study are available in the Gene Expression Omnibus dadabase (https://www.ncbi.nlm.nih.gov/geo/) with the identifier GSE247363.

The source data of this paper are collected in the following database record: biostudies:S-SCDT-10_1038-S44319-024-00291-7.

## Peer review information

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

## Acknowledgements

This research was funded by grants from Beijing Municipal Natural Science Foundation (7222199), Peking University People's Hospital Scientific Research Development Funds (RDX2022-04, RDJP2023-12, RDGS2023-07), and National Natural Science Foundation of China (81971808). We would like to extend our gratitude to Professor Han Wenling from the Department of Immunology, Peking University Health Science Center for generously providing the *Cmtm3* KO and WT mice.

## Author contributions

**Haiyan Xue**: Conceptualization; Data curation; Formal analysis; Funding acquisition; Investigation; Methodology; Writing—original draft; Writing—review and editing. **Ziyan Xiao**: Data curation; Validation; Investigation; Methodology; Writing—original draft; Writing—review and editing. **Xiujuan Zhao**: Resources; Software. **Shu Li**: Resources; Investigation. **Qian Cheng**: Resources; Software; Methodology. **Chun Fu**: Resources; Investigation. **Fengxue Zhu**: Conceptualization; Resources; Software; Supervision; Funding acquisition; Validation; Investigation; Visualization; Project administration.

Source data underlying figure panels in this paper may have individual authorship assigned. Where available, figure panel/source data authorship is listed in the following database record: biostudies:S-SCDT-10_1038-S44319-024-00291-7.

## Disclosure and competing interests statement

The authors declare no competing interests.

# Expanded View Figures

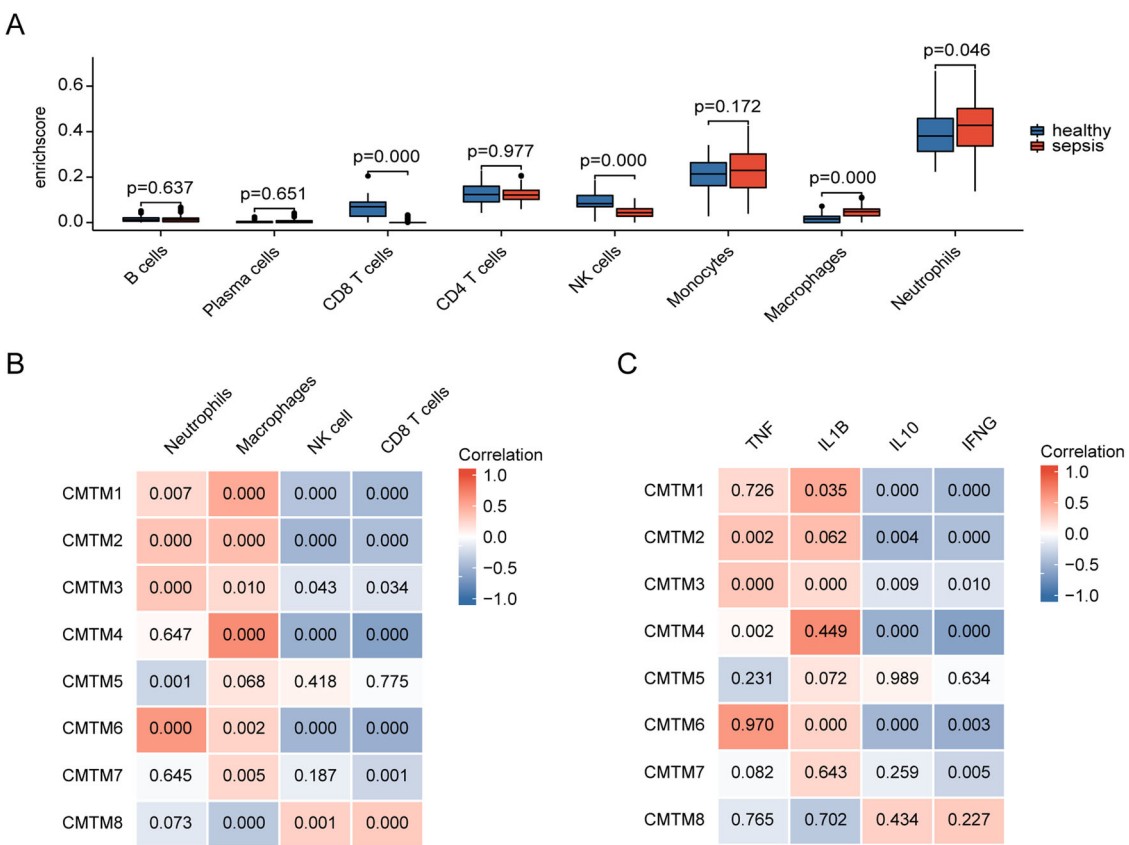

**Figure EV1. Expression and correlation analysis of *CMTM* family in sepsis using public datasets.**

(A) Differences in immune cell abundance between sepsis and healthy groups (healthy = 40, sepsis = 106). (B) Correlation analysis between *CMTM* family members and immune cells (*n* = 146). (C) Correlation analysis between *CMTM* family members and inflammatory cytokines (*n* = 146). Data information: In (A), the boxplot divides the data into quartiles, with the lower and upper edges of the box typically representing the first quartile (Q1) and the third quartile (Q3), respectively. The horizontal line inside the box represents the median, and the whiskers extend to the minimum and maximum values (outliers are indicated by dots, which is the median ± 2 times the interquartile range). In (B, C), the exact *p*-values are provided in the heatmap, with colors representing the range of correlation coefficients.

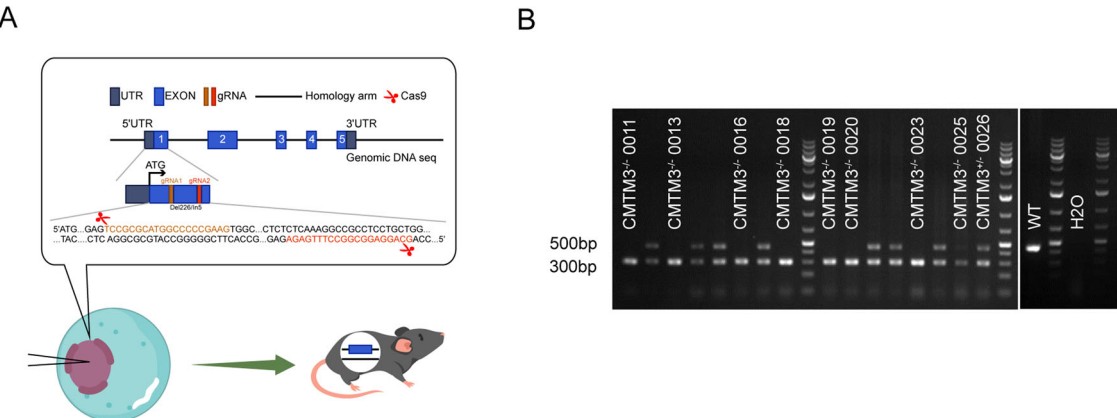

**Figure EV2. Construction and identification of *Cmtm3* KO mice.**

(A) Construction strategy of *Cmtm3* knockout mice. (B) Identification of *Cmtm3* homozygous, heterozygous, and wild-type mice through PCR.

 

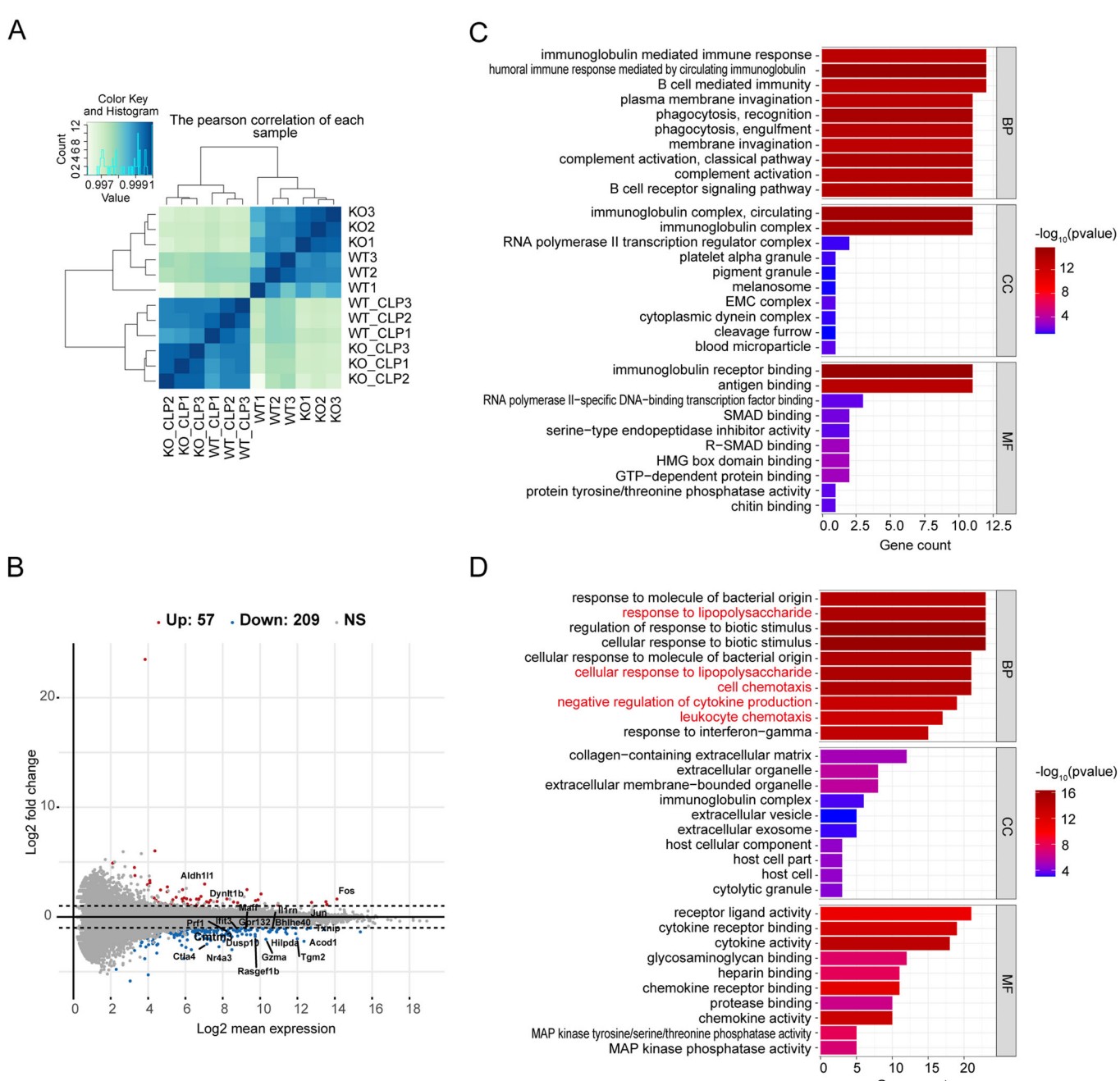

**Figure EV3. RNA-seq analysis of *Cmtm3* KO mice and WT mice.**

(A) Correlation heatmap between pairwise samples. (B) Volcano plot comparing differentially expressed genes between KO CLP and WT CLP (*n* = 3 mice per group, |log₂FoldChange| > 1 and P.adj < 0.05). (C) GO enrichment analysis of upregulated genes comparing KO CLP and WT CLP. (D) GO enrichment analysis of downregulated genes comparing KO CLP and WT CLP. Data information: In (B), data were analyzed by Student's t-est. In (C, D), data were analyzed by Hypergeometric test.

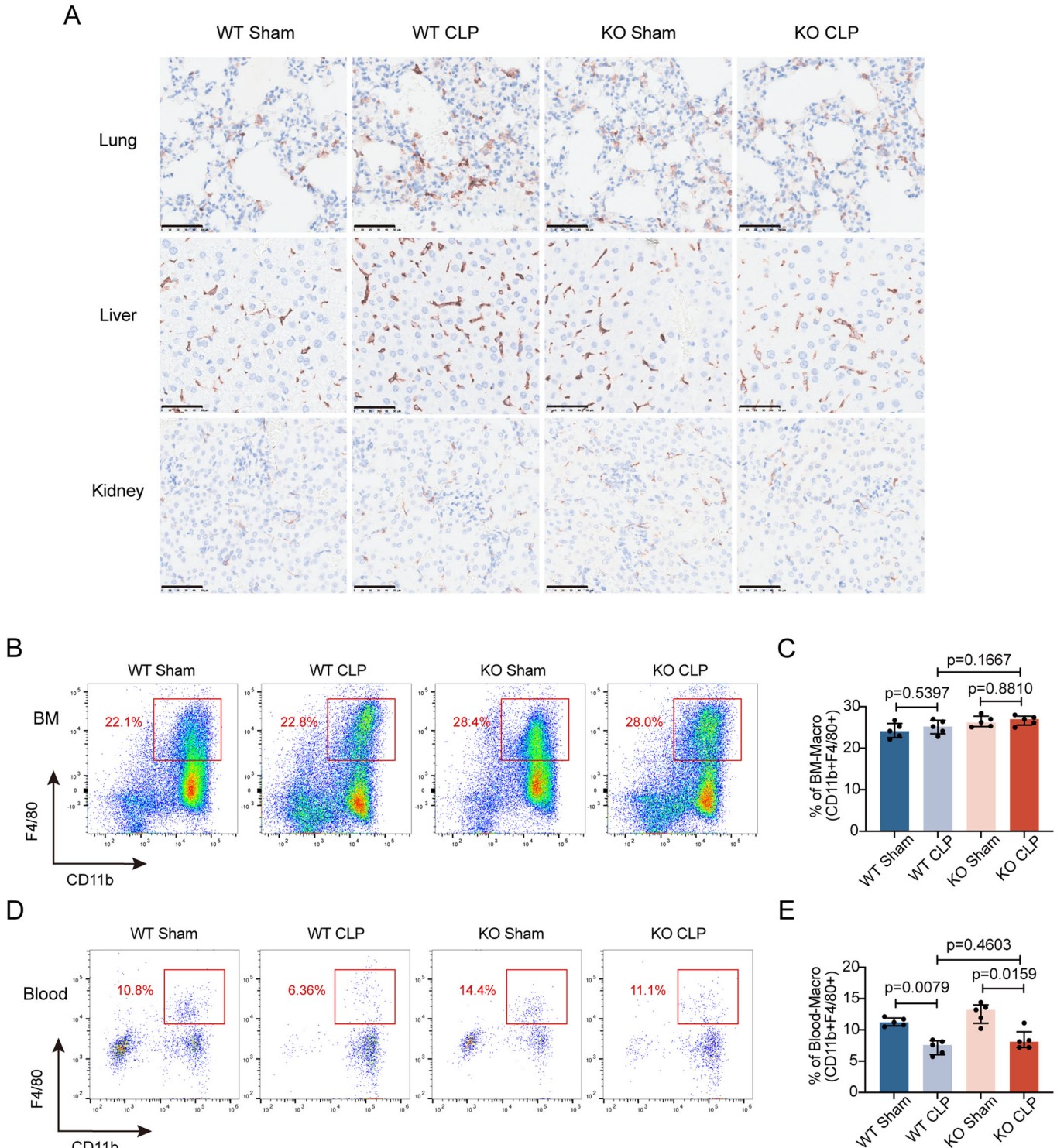

**Figure EV4. The impact of *Cmtm3* KO on the distribution of macrophage populations.**

(A) IHC staining of F4/80 of lung, liver, and kidney 24 h after Sham or CLP (scale bar: 50 μm). (B, C) BM macrophage populations in WT and KO mice 24 h after Sham or CLP (*n* = 5 mice per group). (D, E) Blood macrophage populations in WT and KO mice 24 h after Sham or CLP (*n* = 5 mice per group). Data information: In (C, E), data are expressed as the median ± IQR and were analyzed by Mann–Whitney U test.

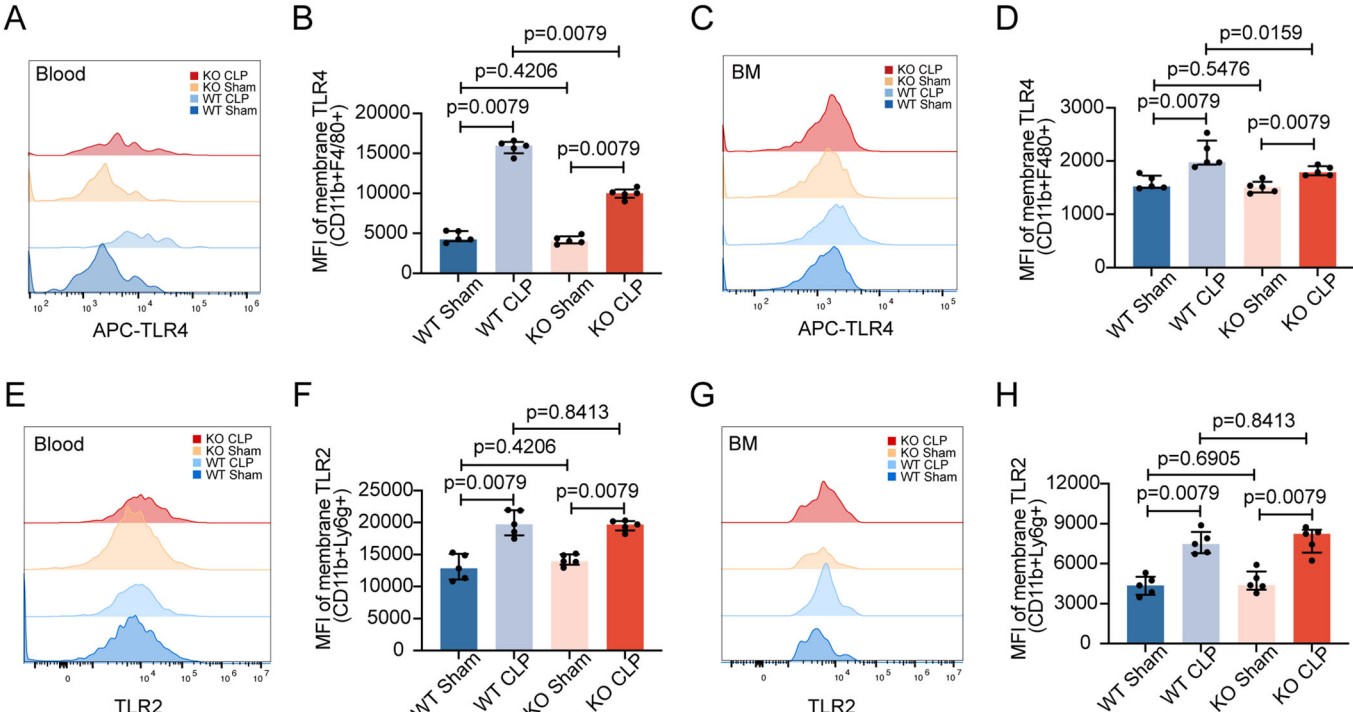

**Figure EV5. The impact of *Cmtm3* KO on TLR4 expression in monocytes and TLR2 expression in neutrophils.**

(A, B) TLR4 membrane expression in blood monocytes in WT and KO mice 24 h after Sham or CLP (*n* = 5 mice per group). (C, D) TLR4 membrane expression in BM monocytes in WT and KO mice 24 h after Sham or CLP (*n* = 5 mice per group). (E, F) TLR2 membrane expression in blood neutrophils in WT and KO mice 24 h after Sham or CLP (*n* = 5 mice per group). (G, H) TLR2 membrane expression in BM neutrophils in WT and KO mice 24 h after Sham or CLP (*n* = 5 mice per group). Data information: In (B, D, F, H), data are expressed as the median ± IQR and were analyzed by Mann–Whitney U test.

