## [Peer Review File · EMBO Reports]

CMTM3 regulates neutrophil activation and aggravates sepsis through TLR4 signaling

Haiyan Xue, Ziyao Xiao, Xiujuan Zhao, Shu Li, Qian Cheng, Chun Fu, and Fengxue Zhu

Corresponding author(s): Fengxue Zhu (zhufengxue@pkuph.edu.cn)

Review Timeline:

Submission Date:	23rd Apr 24
Editorial Decision:	23rd May 24
Revision Received:	22nd Aug 24
Editorial Decision:	6th Sep 24
Revision Received:	30th Sep 24
Accepted:	8th Oct 24

Editor: Achim Breiling / Martina Rembold

Transaction Report:

Dear Prof. Zhu,

Thank you for the submission of your manuscript to EMBO reports. I have now received the reports from the three referees that were asked to evaluate your study, which can be found at the end of this email. As you will see, although the referees feel that the study is of high interest, they have several comments, concerns, and suggestions, indicating that a major revision of the manuscript is necessary to allow publication of the study in EMBO reports. As the reports are below, and all the concerns need to be addressed, I will not detail them here.

Given the constructive referee comments, I would like to invite you to revise your manuscript with the understanding that the concerns of the referees must be addressed in the revised manuscript and in a detailed point-by-point response. Acceptance of your manuscript will depend on a positive outcome of a second round of review. It is EMBO reports policy to allow a single round of revision only and acceptance of the manuscript will therefore depend on the completeness of your responses included in the next, final version of the manuscript.

- 1) a .docx formatted version of the final manuscript text (including legends for main figures, EV figures and tables), but without the figures included. Figure legends should be compiled at the end of the manuscript text.
- 2) individual production quality figure files as .eps, .tif, .jpg (one file per figure), of main figures and EV figures. Please upload these as separate, individual files upon re-submission.

- 4) a complete author checklist, which you can download from our author guidelines

(<https://www.embopress.org/page/journal/14693178/authorguide>). Please insert page numbers in the checklist to indicate where the requested information can be found in the manuscript. The completed author checklist will also be part of the RPF.

- 5) that primary datasets produced in this study (e.g. RNA-seq, ChIP-seq, structural and array data) are deposited in an appropriate public database. If no primary datasets have been deposited, please also state this in a dedicated section (e.g. 'No

primary datasets have been generated and deposited'), see below.

The accession numbers and database should be listed in a formal "Data Availability" section (placed after Materials & Methods) that follows the model below. This is now mandatory (like the COI statement). Please note that the Data Availability Section is restricted to new primary data that are part of this study. This section is mandatory. As indicated above, if no primary datasets have been deposited, please state this in this section

Data availability

8) Regarding data quantification and statistics, please make sure that the number "n" for how many independent experiments were performed, their nature (biological versus technical replicates), the bars and error bars (e.g. SEM, SD) and the test used to calculate p-values is indicated in the respective figure legends (also for EV figures and all those in an Appendix). Please also check that all the p-values are explained in the legend, and that these fit to those shown in the figure. Please provide statistical testing where applicable. Please avoid the phrase 'independent experiment', but clearly state if these were biological or technical replicates. Please also indicate (e.g. with n.s.) if testing was performed, but the differences are not significant. In case n=2, please show the data as separate datapoints without error bars and statistics. See also: <http://www.embopress.org/page/journal/14693178/authorguide#statisticalanalysis>

9) Please add scale bars of similar style and thickness to microscopic images, using clearly visible black or white bars (depending on the background). Please place these in the lower right corner of the images themselves. Please do not write on or near the bars in the image but define the size in the respective figure legend.

10) Please also note our reference format:

12) We now use CRediT to specify the contributions of each author in the journal submission system. CRediT replaces the author contribution section. Please use the free text box to provide more detailed descriptions and do not provide your final manuscript text file with an author contributions section. See also our guide to authors: <https://www.embopress.org/page/journal/14693178/authorguide#authorshipguidelines>

13) We would encourage you to use 'Structured Methods', our new Materials and Methods format. According to this format, the Materials and Methods section should include a Reagents and Tools Table (listing key reagents, experimental models, software,

and relevant equipment and including their sources and relevant identifiers), uploaded as separate file, followed by a Methods and Protocols section in which we encourage the authors to describe their methods using a step-by-step protocol format with bullet points, to facilitate the adoption of the methodologies across labs. More information on how to adhere to this format as well as downloadable templates (.doc or .xls) for the Reagents and Tools Table can be found in our author guidelines (section 'Structured Methods'):

14) Please order the manuscript sections like this, using these names:

Title page - Abstract - Keywords - Introduction - Results - Discussion - Methods - Data availability section - Acknowledgements - Disclosure and Competing Interests Statement - References - Figure legends - Expanded View Figure legends

I look forward to seeing a revised version of your manuscript when it is ready. Please let me know if you have questions or comments regarding the revision.

Yours sincerely,

Referee #1:

This study by Fengxue Zhu et al. describes the role of CMTM3 in sepsis. First the authors find that CMTM3 is upregulated in sepsis patients and correlates (negatively) with disease outcome using publicly available datasets. Next they use CMTM3 KO mice which are protected in the CLP (and LPS model) by regulating neutrophil activation through CXCR2 expression. This protection is mediated via reducing TLR4 expression and TLR4 overexpression prevents the protective effect of CMTM3KO.

I have some major concerns:

1. Biggest concern is that the authors compare (in house bred) cmtm3KO mice with SPF mice from a separate breeding. Especially when doing CLP it is necessary to compare genotypes derived from littermates since different breedings / origins of mice can have a huge impact on the microbiome and immunity of mice. I would propose to make a cross of your wt and KO mice, to obtain a heterozygous breeding (F1) that can then be used to obtain your littermate controls (F2 breeding). See Robertson et al. Cell Reports 2019 for this issue.
2. The authors use systemic cmtm3 knock out mice, but focus on the role of CMTM3 on the neutrophils. To really pinpoint the role of CMTM3 on neutrophils, it is needed to make a conditional knock out. As an alternative the authors can make use of 1A8 antibody to deplete neutrophils and observe whether the KO mice are no longer protected against CLP.
3. As neutrophils can no longer mobilize in the KO mice, it might have an influence on the bacterial clearance in the CLP model. Do the KO mice have more bacteria in blood and other organs?
4. The authors see lower CMTM3 expression in the sepsis non-survivors, but the KO mice have a reduced lethality. This feels counterintuitive and should be better discussed in discussion part.
5. A really important experiment is Fig 7 where the authors overexpress TLR4 in cmtm3 ko mice thereby loosening their phenotype in the LPS model. But this is only done in the LPS model and not in the CLP model. The LPS model is actually not a good model for studying sepsis, and this should be done in the CLP model too. A real advantage of this overexpression is that via this way littermate controls are used.

Minor comments:

1. The authors used publicly available datasets to compare CMTM expression with survival information. But which kind of datasets are these? PBMCs? Whole blood? Please specify.
2. Specify unit in correlation curves with immune cells
3. Specify value and type of correlation used per dataset in legend
4. Figure 5i-j: 50 is 50ng/ml LPS? Be more clear. Moreover, how do you explain the higher migration cell number in LPS - group comparing the 50 and 0 subgroups? As both should have no LPS?
5. TLR4KO mice have been described multiple times in literature in the CLP model, but this gives different outcomes. How do the authors explain this disparity?
6. The title is confusing of paragraph for figure 7: TLR4 overexpression does not rescue the mice, but sensitises the CMTM3KO mice for LPS.
7. Figure 7h: colors are not clear. Use a thicker lining.
8. Specify broad spectrum antibiotics used and volume.
9. Specify dose of LV-TLR4 (not only volume)
10. Scr => sCr to be more clear that this is creatinine

Referee #2:

This is an interesting report based on the eloquent series of experiments to explore the role of CMTM3 in leukocytes in sepsis. The study is based on an analysis of differential gene expression in publicly available transcriptomic databases from sepsis patients. The authors focus on the CMTM family because these interesting molecules have not been studied in sepsis models previously. They identify a previously unknown role for CMTM3 in regulating neutrophil surface TLR4 expression on immature neutrophils in the bone marrow and link this to the egress of PMN from the bone marrow to the circulation and into organs. Blocking this through the use of knockouts prevents the egress of PMN and improves survival in a CLP model in male mice treated with antibiotics. While the results are notable and potentially of interest to the sepsis field, there are gaps in the work that should be addressed. These are listed as follows:

1. Clearly the CMTM3-TLR4 axis has been conserved in biology for its survival benefit. In the model used in this paper the authors chose a CLP model followed by antibiotics. Presumably, the egress of PMN would be important for the clearance of bacteria and the antimicrobial effector functions of the immune system. The CLP model should be performed without the use of antibiotics to determine whether the inhibition of neutrophil egress impairs microbial clearance functions in the absence of antibiotics. In addition, the authors should expand their discussion to discuss the adaptive functions of this pathway and how it then can become a feature of the pathogenesis of sepsis.
2. The authors chose to use only male mice, however, provide no rationale for not including female mice. The lack of female mice should be mentioned as an important limitation. Can they speculate on how female mice would respond to CMTM3 deletion?
3. TLR4 has major functions on a number of other important immune cell types, especially monocytes and macrophages. Did the deletion of CMTM3 change TLR4 expression on monocytes or macrophages.
4. Is the effect of CMTM3 specific to TLR4? While the informatics point strongly to TLR4 signaling as a dominant mechanism other surface TLR clearly play a role in polymicrobial infection such as seen in CLP. Therefore, providing data on another TLR4 (such as TLR2) would address the issue of selectivity.
5. What criteria were used to select the 106 patients used for analysis?
6. The authors should provide a description of the data types (bulk RNAseq?) that were used for the analysis of sepsis patients. For example, were these transcriptomic datasets using whole blood or leukocyte populations from the circulation?
7. As a minor point it would be helpful if the figures were numbered.

Referee #3:

The manuscript reports on a novel finding indicating a previously unrecognized involvement of CMTM3 in leukocyte responses during sepsis.

Multiple approaches were employed in sequential experiments that led to author's conclusions.

I have following questions/comments to the authors:

- *Fig. 1f - data on neutrophils and macrophages in septic patients - is this data for the whole body or some particular tissues?
- *the differences between WT and KO mice are clear and statistically significant yet not dramatic. Therefore terms such as "pivotal role" (e.g. line 418) should be avoided. Clearly other molecules are also involved in TLR4 and CXCL2 regulation. I agree that finding out the precise mechanism would require the whole new study but some speculations would be of value.
- *in regard to the above and bioinformatic data - any other molecules show correlation with TLR4 and/or CXCL2 during sepsis?

Minor:

- figures should be numbered as embed in the manuscript I difficult to follow.
- Ly6G - with a capital G.
- LPS-induced sepsis - change sepsis to endotoxemia.
- morphology of various organs is only shown in images; a scoring system would allow for quantification of these data.
- Discussion could be shortened

Referee #1:

This study by Fengxue Zhu et al. describes the role of CMTM3 in sepsis. First the authors find that CMTM3 is upregulated in sepsis patients and correlates (negatively) with disease outcome using publicly available datasets. Next they use CMTM3 KO mice which are protected in the CLP (and LPS model) by regulating neutrophil activation through CXCR2 expression. This protection is mediated via reducing TLR4 expression and TLR4 overexpression prevents the protective effect of CMTM3KO.

Major concerns:

1. Comment: Biggest concern is that the authors compare (in house bred) cmtm3KO mice with SPF mice from a separate breeding. Especially when doing CLP it is necessary to compare genotypes derived from littermates since different breedings / origins of mice can have a huge impact on the microbiome and immunity of mice. I would propose to make a cross of your wt and KO mice, to obtain a heterozygous breeding (F1) that can then be used to obtain your littermate controls (F2 breeding). See Robertson et al. Cell Reports 2019 for this issue.

Response: We are deeply grateful for your insightful and attentive observations regarding the genetic background and microbiome consistency of the mice used in our study. We have taken your suggestions very seriously and have thoroughly reviewed our animal sourcing and breeding procedures. Upon re-evaluation, we confirm that the parental Cmtm3 KO mice and their WT counterparts, sharing the same genetic background, were provided by Professor Han Wenling from the Peking University Health Science Center. We have verified the breeding history of these mice and have determined that both KO and WT mice were bred under the same conditions. After the construction of the Cmtm3 KO mice, to address potential off-target mutations that could arise from guideRNA, the F1 generation of heterozygous mice was backcrossed with WT C57BL/6 mice for six generations. Subsequently, homozygous Cmtm3 KO and WT mice were obtained through interbreeding of the F7 generation. All subsequent mice were derived from these homozygous pairs through inbreeding. We believe that this rigorous breeding protocol ensures that the genetic background and environmental conditions of the WT and KO mice are indeed identical, thereby minimizing variability that could affect the microbiome and immune status of the mice.

We sincerely apologize for not providing sufficient description in the previous manuscript to elucidate this critical information. To clarify any potential misunderstandings from our previous methods description, we have revised the section pertaining to the animal source and methodology as follows:

“Ten-week-old parental Cmtm3 gene knockout (KO) mice and their wild-type (WT) counterparts, sharing the same genetic background, were sourced from Professor Han Wenling's laboratory at the Department of Immunology, Peking University Health Science Center. These mice were originally of the same genetic background and were subjected to a controlled breeding program to ensure genetic consistency. All mice were housed in the Animal Center of Peking University People's Hospital under specific pathogen-free (SPF) conditions. Mice aged 8 to 10 weeks, with body weights ranging from 18 to 20 grams, were utilized for the experiments. Every effort was made to ensure age and weight matching among the mice in the experimental groups.”

We hope this revised description addresses your concerns and provides a clear understanding of

our experimental design and animal sourcing. Thank you once again for your valuable feedback.

2. Comment: The authors use systemic *cmtm3* knock out mice, but focus on the role of CMTM3 on the neutrophils. To really pinpoint the role of CMTM3 on neutrophils, it is needed to make a conditional knock out. As an alternative the authors can make use of 1A8 antibody to deplete neutrophils and observe whether the KO mice are no longer protected against CLP.

Response: We appreciate your thoughtful suggestions regarding the role of CMTM3 in neutrophils and the potential impact of other cell types on the observed effects. Your feedback are valuable in guiding our revisions.

Our current data, including bioinformatics analysis, immunohistochemistry, and flow cytometry with neutrophil-specific markers, consistently point to a significant role for neutrophils in the biological effects related to CMTM3. This evidence strongly supports the involvement of neutrophils in CMTM3 function. While neutrophils seem to be the primary cells of interest, we recognize that without a cell type-specific knockout model, we cannot definitively rule out the contributions of other cell types. We understand the importance of further studies using conditional knockout mice to clarify CMTM3's role in neutrophils. However, due to our current experimental limitations, we are unable to perform these experiments at this time.

As an alternative, we conducted experiments using the 1A8 antibody to deplete neutrophils, as shown in survival rate figure provided below. Our findings indicate that depleting neutrophils reduced the protective effect seen in *Cmtm3* knockout mice. However, we also observed a higher mortality rate in wild-type septic mice treated with the 1A8 antibody. We suspect that the higher mortality rate observed in wild-type septic mice after 1A8 antibody treatment could be attributed to the impaired pathogen clearance following neutrophil depletion. We believe that the protective effect of *Cmtm3* knockout might be related to its influence on the abnormal migration of neutrophils, rather than their total removal. This makes it challenging to prove the specificity of CMTM3's protective effect using the current approach.

We have expanded the discussion section to include these limitations and their impact on interpreting our results. Although our data do not exclude the influence of CMTM3 on other cells, they do support a pivotal role for CMTM3 in neutrophils. We acknowledge that exploring the roles of other cell types is an important direction for future research, as clearly stated in our revisions.

We hope that these revisions and clarifications address your concerns and strengthen the

conclusions of our study. We are grateful for the opportunity to improve our work with your expert feedback and are confident that our research provides useful insights into CMTM3's role in sepsis.

3. Comment: As neutrophils can no longer mobilize in the KO mice, it might have an influence on the bacterial clearance in the CLP model. Do the KO mice have more bacteria in blood and other organs?

Response: Thank you for your thoughtful comment. We acknowledge the importance of assessing the role of neutrophil mobilization in bacterial clearance within the context of the CLP model of sepsis.

In our preliminary experiments, we indeed evaluated the bacterial burden in both WT and Cmtm3 KO mice following the CLP procedure. Our initial findings indicated that the absence of Cmtm3 led to a compromised bacterial clearance capability, resulting in an increased colony count in the peritoneal lavage fluid of the KO mice compared to the WT mice. Recognizing the potential impact of this finding on the protective effect of Cmtm3 knockout in CLP mice, we subsequently administered antibiotics post-CLP in our experiments. Upon re-evaluation with antibiotic treatment, we observed that the bacterial counts in the peritoneal lavage fluid between the WT and KO mice were no longer significantly different. This suggests that the antibiotic intervention effectively mitigated the adverse effects of reduced neutrophil mobilization in the KO mice.

We have incorporated these insights into the revised manuscript to provide a more comprehensive understanding of the role of Cmtm3 in the immune response to sepsis and the potential benefits of antibiotic treatment. We are grateful for the opportunity to delve deeper into this aspect of our research and for your constructive feedback.

4. Comment: The authors see lower CMTM3 expression in the sepsis non-survivors, but the KO mice have a reduced lethality. This feels counterintuitive and should be better discussed in discussion part.

Response: Thank you for raising this important issue. We had indeed noticed the peculiarity of these results and conducted rigorous checks on the accuracy and reproducibility of our experiments and findings. After thorough validation, we confirmed that the expression of CMTM3 is indeed lower in non-survivors compared to survivors, and the KO of Cmtm3 does improve the survival rate in septic mice. We have addressed this counterintuitive observation in our discussion with the following points:

- i) Complexity of sepsis immune regulation: The immune response in sepsis is complex and multifaceted, rather than singular and sequential. Both the innate and adaptive immune systems are activated during sepsis, with a coexistence of excessive immune responses and immune suppression. We believe that the more severe immune suppression in non-survivors leads to a reduced capacity for immune synthesis, resulting in lower CMTM3 expression. In contrast, the higher expression of CMTM3 in survivors may represent an adaptive immune response.
- ii) Gene expression and survival outcomes: The relationship between gene expression and survival outcomes is not always positively linear, especially in a complex syndrome like sepsis. Correlations in gene expression can only serve as clues, guiding the research into the gene's function. However, to definitively determine whether a gene is pathogenic or protective, the gold

standard should be the survival rate of genetically edited mice.

iii) Potential multicellular role of CMTM3: We do not exclude the possibility that CMTM3 plays a role in multiple cell types. Therefore, further experiments are needed to clarify the specific mechanisms of CMTM3's action in sepsis and its relationship with host survival rates.

We have expanded our discussion to provide a more comprehensive explanation of these findings and their implications for understanding the pathophysiology of sepsis. Thank you for the opportunity to clarify and enhance our manuscript.

5. Comment: A really important experiment is Fig 7 where the authors overexpress TLR4 in *cmtm3* ko mice thereby loosening their phenotype in the LPS model. But this is only done in the LPS model and not in the CLP model. The LPS model is actually not a good model for studying sepsis, and this should be done in the CLP model too. A real advantage of this overexpression is that via this way littermate controls are used.

Dear Reviewer,

Response: Thank you for your insightful comments and suggestions regarding the importance of the experiment depicted in Figure 7. We concur with your assessment that the LPS model has limitations when used as the sole representative of sepsis studies. In response to your constructive feedback, we have conducted additional experiments in the more clinically relevant CLP model (**Figure 8**). We are pleased to report that the results from the CLP model corroborate our findings from the LPS model, further reinforcing the robustness of our conclusions. We have incorporated these new data into the revised manuscript, which we believe addresses your concerns and contributes to a more comprehensive understanding of the role of CMTM3 in sepsis pathophysiology. Thank you once again for your valuable input, which has undoubtedly enriched our study.

Minor comments:

1. Comment: The authors used publicly available datasets to compare CMTM expression with survival information. But which kind of datasets are these? PBMCs? Whole blood? Please specify.

Response: We utilized whole blood datasets that are publicly available. This specific detail has been clarified and indicated in the revised manuscript. Thank you for your attention to this detail.

2. Comment: Specify unit in correlation curves with immune cells

Response: Thank you. We would like to clarify that the infiltration status of immune cells was analyzed using the CIBERSORT algorithm. This analysis is based on gene expression profile data, which is consistent with the type of gene expression profile data for the CMTM family. Therefore, the correlation curves generated by CIBERSORT do not have specific units, as they represent the relative proportions of different immune cell types derived from the expression profiles. We hope this explanation addresses your concern and provides the necessary clarity on the methodology used in our study.

3. Comment: Specify value and type of correlation used per dataset in legend

Response: Thank you. We have made the necessary revisions in the manuscript to specify the value and type of correlation used for each dataset in the legend.

4. Comment: Figure 5i-j: 50 is 50ng/ml LPS? Be more clear. Moreover, how do you explain the higher migration cell number in LPS - group comparing the 50 and 0 subgroups? As both should have no LPS?

Response: Thank you for pointing out this. In Figure 5i-j, the term "50" indeed refers to the concentration of CXCL2 in ng/ml, specifically 50 ng/ml, used in the transwell migration assay. We acknowledge that the previous manuscript was not explicit about this, and we appreciate your observation. The discrepancy in the number of migrated cells between the 50 and 0 subgroups can be attributed to the presence of different concentrations of CXCL2, which is a potent chemoattractant for neutrophils expressing the CXCR2 receptor. The higher number of migrated cells in the 50 subgroup as compared to the 0 subgroup is likely due to the chemotactic gradient established by the presence of CXCL2. We apologize for any confusion caused by the lack of clarity in our initial submission. In the revised manuscript, we have included a clear explanation of the concentrations used and the role of CXCL2 in the migration assay. The figure has been updated to reflect this information, ensuring that readers have a comprehensive understanding of the experimental conditions and results.

5. Comment: TLR4KO mice have been described multiple times in literature in the CLP model, but this gives different outcomes. How do the authors explain this disparity?

Response: We appreciate the opportunity to address the important aspect of our study regarding the varied outcomes associated with TLR4 KO mice in the CLP model as reported in the literature. Sepsis is a complex syndrome with a multifaceted pathophysiology that involves both the immune system's efforts to control infection and the dysregulated activation of immune responses leading to inflammation and immunosuppression. TLR4, implicated in these processes, may manifest different roles depending on the phase of sepsis, the severity of the condition, and the specific disease state. The timing and nature of interventions and assessments may thus yield divergent results.

It is important to note that while sepsis often exhibits overactivation of the TLR4 receptor, leading to excessive inflammatory responses, TLR4-mutant mice struggle to control low-dose gram-negative infections, indicating the necessity for precise management of TLR4 expression during infection rather than its outright deletion. The molecular regulatory mechanisms controlling TLR4 expression are therefore of significant interest and are essential for the precise modulation of TLR4 in sepsis.

Moreover, disparities in the literature could be attributed to differences in experimental conditions, such as the genetic background of the TLR4 KO mice, the choice of assessment criteria, and the timing of experiments. These factors may all influence the outcomes and underscore the need for a nuanced analysis of each experimental context, especially for a complex syndrome like sepsis.

In our study, we have taken these factors into consideration and have endeavored to provide a comprehensive analysis that reflects the intricate nature of sepsis. We have also discussed these points in the manuscript to ensure that our findings are interpreted within the broader context of existing literature and the complexities of the disease. Understanding the molecular mechanisms that govern TLR4 expression is vital for the development of targeted therapies that can fine-tune TLR4 activity, potentially offering a more effective approach to sepsis management.

Thank you again for the inspiring question, which has allowed us to delve deeper into the

nuances of TLR4's role in sepsis and the importance of its precise regulation.

6. Comment: The title is confusing of paragraph for figure 7: TLR4 overexpression does not rescue the mice, but sensitises the CMTM3KO mice for LPS.

Response: Thank you for your guidance on the title for Figure 7. We have taken your suggestion into account and have revised the title to better reflect the findings of our study. The updated title now is "Overexpression of TLR4 Reverses the Protective Effect of Cmtm3 KO in LPS-Induced Endotoxemia." We are grateful for the opportunity to enhance the clarity of our manuscript.

7. Comment: Figure 7h: colors are not clear. Use a thicker lining.

Response: Thank you for the feedback on Figure 7h. We have revised the figure with bolder lines for better clarity.

8. Comment: Specify broad spectrum antibiotics used and volume.

Response: Thank you for your comment. We have revised the manuscript to specify that the broad-spectrum antibiotic used in our study was Ampicillin, administered at a dosage of 50 milligrams per kilogram, every 12 hours. This detail has been added to the methods section for clarity.

9. Comment: Specify dose of LV-TLR4 (not only volume)

Response: Thank you for your comment. We have now specified the dose of LV-TLR4 in our revised manuscript. The dose used was 10^8 infectious units (IFU). This detail has been added to the methods section to provide clarity and precision regarding the experimental conditions.

10. Comment: Scr => sCr to be more clear that this is creatinine

Response: Thank you for pointing out the clarification needed. We have updated the manuscript to use "sCr" to clearly indicate serum creatinine.

Referee #2:

This is an interesting report based on the eloquent series of experiments to explore the role of CMTM3 in leukocytes in sepsis. The study is based on an analysis of differential gene expression in publicly available transcriptomic databases from sepsis patients. The authors focus on the CMTM family because these interesting molecules have not been studied in sepsis models previously. They identify a previously unknown role for CMTM3 in regulating neutrophil surface TLR4 expression on immature neutrophils in the bone marrow and link this to the egress of PMN from the bone marrow to the circulation and into organs. Blocking this through the use of knockouts prevents the egress of PMN and improves survival in a CLP model in male mice treated with antibiotics. While the results are notable and potentially of interest to the sepsis field, there are gaps in the work that should be addressed. These are listed as follows:

1. Comment: Clearly the CMTM3-TLR4 axis has been conserved in biology for its survival benefit.

In the model used in this paper the authors chose a CLP model followed by antibiotics. Presumably, the egress of PMN would be important for the clearance of bacteria and the antimicrobial effector functions of the immune system. The CLP model should be performed without the use of antibiotics to determine whether the inhibition of neutrophil egress impairs microbial clearance functions in the absence of antibiotics. In addition, the authors should expand their discussion to discuss the adaptive functions of this pathway and how it then can become a feature of the pathogenesis of sepsis.

Response: We are grateful for the your constructive feedback and focus on the importance of the CMTM3-TLR4 axis in the immune response to sepsis. The suggestions to perform the CLP model without antibiotics and to expand the discussion on the adaptive functions of this pathway are highly relevant to our study.

In response to the first point, we have indeed conducted preliminary experiments to assess the impact of Cmtm3 knockout on bacterial clearance in the CLP model. Our initial findings indicated that the absence of Cmtm3 led to an increased bacterial burden in the peritoneal lavage fluid of the KO mice, suggesting that the egress of PMNs is crucial for effective microbial clearance. Considering the potential implications of this finding on the protective role of Cmtm3 knockout in the CLP model, we proceeded to administer antibiotics post-CLP to determine if this intervention could mitigate the observed impairment in bacterial clearance. Upon re-evaluation with antibiotic treatment, we found that the bacterial counts in the peritoneal lavage fluid between the wild-type and knockout groups were no longer significantly different, indicating that the antibiotic treatment effectively compensated for the reduced neutrophil egress in the KO mice. Therefore, in our study, the primary model used was the CLP model followed by antibiotic administration. To address the second point, we have expanded our discussion in the revised manuscript to include the adaptive functions of the CMTM3-TLR4 pathway and its potential role in the pathogenesis of sepsis. We now discuss how the inflammatory responses balance modulated by the CMTM3-TLR4 axis is critical for the resolution of infection and the prevention of tissue damage.

We appreciate the opportunity to refine our study and provide a more comprehensive analysis of the CMTM3-TLR4 axis in the context of sepsis. Your feedback has been invaluable in enhancing the depth and relevance of our research. Thank you for your insightful comments.

2. Comment: The authors chose to use only male mice, however, provide no rationale for not including female mice. The lack of female mice should be mentioned as an important limitation. Can they speculate on how female mice would respond to CMTM3 deletion?

Response: Thank you for your comment regarding the use of only male mice in our study. We appreciate your insight and would like to address the concerns raised.

In our study, the decision to utilize only male mice was based on two primary considerations:

- i) Stability of hormone levels: Male mice exhibit relatively stable hormone levels, whereas the hormonal profile in female mice fluctuates throughout their estrous cycle. These fluctuations could potentially affect the stability and reproducibility of the experimental outcomes. Given the complex interplay between hormones and immune responses, the additional variability introduced by the estrous cycle in female mice could complicate the interpretation of the results related to immune function and disease pathophysiology.
- ii) Convenience of experimental design: Male mice typically have a larger body size, which

facilitates surgical procedures and the measurement of physiological parameters. The ease of handling and the reduced likelihood of procedural complications in male mice contribute to the reliability of our experimental design, especially when invasive techniques are involved.

We acknowledge that the exclusion of female mice represents a limitation in our study. It is possible that the response to *Cmtm3* deletion in female mice could differ due to the aforementioned hormonal influences on immune function. Future studies will need to investigate the impact of sex differences to fully elucidate the role of CMTM3 in both male and female mice. We have included a discussion on this limitation in the revised manuscript and will consider the inclusion of female mice in our future studies to provide a more comprehensive understanding of the effects of *Cmtm3* deletion. Thank you for providing us with the opportunity to address this important aspect of experimental design.

3. Comment: TLR4 has major functions on a number of other important immune cell types, especially monocytes and macrophages. Did the deletion of CMTM3 change TLR4 expression on monocytes or macrophages.

Response: Thank you for your insightful comment and for prompting us to investigate the impact of CMTM3 deletion on TLR4 expression in monocytes and macrophages. We have conducted additional experiments to address this question, and the results are now included as **Figure EV5A-D** in the revised manuscript.

Our supplementary experiments reveal that the deletion of CMTM3 indeed alters TLR4 expression on both peripheral blood and bone marrow monocytes. Post-CLP model induction, we observed an increase in TLR4 expression on the surface of monocytes from WT mice. In contrast, the KO mice exhibited a reduced increment in TLR4 expression on monocytes following the CLP procedure. These new data provide further insights into how CMTM3 may modulate TLR4 expression, particularly in the context of sepsis, and we believe they add significant value to our study. We have incorporated these findings into the revised manuscript to ensure a thorough understanding of CMTM3's role in immune cell function.

We are grateful for the opportunity to enhance our research with your expert feedback and are confident that these additional experiments contribute to the depth and relevance of our investigation.

4. Comment: Is the effect of CMTM3 specific to TLR4? While the informatics point strongly to TLR4 signaling as a dominant mechanism other surface TLR clearly play a role in polymicrobial infection such as seen in CLP. Therefore, providing data on another TLR4 (such as TLR2) would address the issue of selectivity.

Response: Thank you for your insightful question regarding the specificity of CMTM3's effect on TLR4 in comparison to other surface TLRs, particularly in the context of polymicrobial infection as seen in the CLP model.

To address this concern, we conducted additional experiments to evaluate the impact of CMTM3 deletion on TLR2 expression. The results of these experiments are now presented as **Figure EV5E-H** in the revised manuscript. Our findings indicate that the deletion of CMTM3 does not affect the expression of TLR2 on the surface of peripheral blood and bone marrow neutrophils. Following the CLP model induction, we observed an increase in TLR2 expression on neutrophils from WT mice, which is consistent with the known inflammatory response. Notably, the KO mice

showed no difference in the increment of TLR2 expression on neutrophils post-CLP, suggesting that the effect of CMTM3 deletion is specific to TLR4 and not a general effect on all TLRs. These results support the notion that CMTM3's influence is selective for TLR4 and provide further evidence for the specificity of its role in modulating immune responses during sepsis. We have incorporated these findings into the revised manuscript to contribute to a more nuanced understanding of CMTM3's function.

We appreciate your suggestion to explore the selectivity of CMTM3's effects and are pleased to have been able to include this additional layer of analysis in our study.

5. Comment: What criteria were used to select the 106 patients used for analysis?

Response: Thank you for your insightful question regarding the selection criteria for the 106 patients included in our analysis. Our study utilized a systematic approach to identify relevant datasets within the GEO database. We conducted a comprehensive search using the keyword "sepsis" to ensure a broad and inclusive retrieval of potential datasets. The datasets were meticulously reviewed, and those that featured survival information of sepsis patients, as well as contained expression data of the CMTM family of genes, were selected for further analysis. The 106 patients in our study represent a deduplicated count of sepsis patients from the datasets that met these specific inclusion criteria. This process was designed to ensure that our analysis was based on a robust and clinically relevant patient cohort, allowing for a more accurate interpretation of the results. We hope this provides a clear explanation of our patient selection process.

6. Comment: The authors should provide a description of the data types (bulk RNAseq?) that were used for the analysis of sepsis patients. For example, were these transcriptomic datasets using whole blood or leukocyte populations from the circulation?

Response: Thank you for pointing out this. The datasets GSE33118, GSE54514, and GSE95233, which were included in our analysis, consist of bulk RNA sequencing data derived from peripheral whole blood samples of sepsis patients. This specific detail has been clarified and indicated in the revised manuscript. Thank you for your attention to this detail.

7. Comment: As a minor point it would be helpful if the figures were numbered.

Response: Thank you for your comment. We will ensure that the figures are numbered in the revised manuscript. We appreciate the opportunity to refine our work based on your feedback.

Referee #3:

The manuscript reports on a novel finding indicating a previously unrecognized involvement of CMTM3 in leukocyte responses during sepsis.

Multiple approaches were employed in sequential experiments that led to author's conclusions.

I have following questions/comments to the authors:

Major:

1. Comment: Fig. 1f - data on neutrophils and macrophages in septic patients - is this data for the whole body or some particular tissues?

Response: Thank you for your inquiry regarding Fig. 1f. The data presented on neutrophils and macrophages in septic patients were obtained using the CIBERSORT algorithm, which analyzed the RNA sequencing data derived from peripheral whole blood samples. This approach provides an estimation of the infiltration status of these immune cells in the systemic circulation rather than specific tissue samples. We hope this clarifies the source and nature of the data presented in our figure.

2. Comment: the differences between WT and KO mice are clear and statistically significant yet not dramatic. Therefore terms such as "pivotal role" (e.g. line 418) should be avoided. Clearly other molecules are also involved in TLR4 and CXCL2 regulation. I agree that finding out the precise mechanism would require the whole new study but some speculations would be of value.

Response: Thank you for your insightful feedback on our manuscript. We appreciate your observation regarding the use of the term "pivotal role" in the context of our findings on CMTM3's involvement in sepsis. In light of your comments, we have revised the words to "potential contributor" to better reflect the nuances of our results. Furthermore, regarding the precise mechanisms of CMTM3's involvement, we would like to offer some informed speculation based on the known functions of the CMTM family. Typically acting as auxiliary molecules for key proteins, we hypothesize that CMTM3 may serve as a facilitator for TLR4, potentially stabilizing its expression on the cell membrane. This hypothesis is currently under active investigation, and we acknowledge that substantial work will be required to validate or refute this conjecture. We appreciate the opportunity to refine our manuscript with your guidance and look forward to contributing further to the understanding of CMTM3's role in sepsis. Thank you once again for your valuable input.

3. Comment: in regard to the above and bioinformatic data - any other molecules show correlation with TLR4 and/or CXCL2 during sepsis?

Response: Thank you for your inquiry about additional molecules correlated with TLR4 and/or CXCL2 during sepsis. In our bioinformatic analysis, as depicted in Figure 4a, we identified the top 20 molecules with the highest correlation, which included MAPK3, GRB2, RAC1, LYN, TLR4, NFKB1, and MYD88, among others. Our findings indicate that a significant number of these molecules are part of the TLR4 signaling pathway, underscoring the central role of this pathway in the sepsis response. We hope this information provides the necessary detail to address your comment.

Minor:

1. Comment: figures should be numbered as embed in the manuscript I difficult to follow.

Response: Thank you for your comment. We will ensure that the figures are numbered in the revised manuscript.

2. Comment: Ly6G - with a capital G.

Response: We apologize for the oversight. We have made the correction and ensure that "Ly6G" is consistently formatted with a capital 'G' throughout the manuscript. Thank you for your

attention to detail.

3. Comment: LPS-induced sepsis - change sepsis to endotoxemia.

Response: Thank you for your comment. We understand the distinction and have revised the manuscript accordingly to use "endotoxemia" in the revised manuscript.

4. Comment: morphology of various organs is only shown in images; a scoring system would allow for quantification of these data.

Response: Thank you for your constructive comment. We agree that incorporating a scoring system would provide a more quantitative assessment of the organ morphology data. We have revised the manuscript to include such a system as per your suggestion.

5. Comment: Discussion could be shortened

Response: Thank you for your suggestion. We have carefully reviewed the content and have made the necessary revisions to streamline the discussion, removing redundant language and refining our points for clarity and conciseness. We believe these changes enhance the overall readability and focus of the paper. We appreciate the opportunity to refine our work based on your feedback.

Dear Prof. Zhu

Thank you for the submission of your revised manuscript to EMBO reports. As my colleague Achim Breiling is currently not in the office, I have temporarily taken over the handling of your manuscript.

We have now received the full set of referee reports that is copied below. As you will see, all referees are very positive about the study and request only minor changes to clarify text and figures.

From the editorial side, there are also a few things that we need before we can proceed with the official acceptance of your study.

- Source Data: At EMBO Press, we ask the authors to provide the source data that were used to generate the main figures. You should have received an e-mail from our Source Data Coordinator Hannah Sonntag on May 27, 2024, detailing all the source data we would need. Please check your mailbox (and spam folder) whether you have received this e-mail. I have also re-sent it now. Please contact us in case none of these e-mails came through.

- Please update the 'Conflict of interest' paragraph to our new 'Disclosure and competing interests statement'. For more information see

<https://www.embopress.org/page/journal/14693178/authorguide#conflictsofinterest>

- References need to be alphabetical, not numerical; et al needs to be used after 10 author names; DOIs should only be used for preprints and datasets that have not been published yet.

- Author Checklist

1) Experimental study design and statistics: you indicate that you included statements about sample size, randomization, blinding, inclusion/exclusion criteria in the Materials and Methods section of the manuscript, but such statements seem to be missing. Please either include this information in the Methods or choose 'Not applicable' instead of 'Yes' in the pull-down menu of the Checklist.

2) Experimental animals: You state in the Author Checklist that you used 'Animals observed in or captured from the field'. As far as I could see, your experiments did not include field work or animals sampled from the wild. Please check and correct this entry.

3) You indicate in the Author Checklist that you deposited 'human clinical and genomic datasets in a public access-controlled repository'. As far as I can see, you refer to already published datasets that you re-used in the Data availability section (in addition to your own dataset GSE247363). Since these datasets from human patients were not generated by you in this study, you can choose 'Not applicable' here.

4) Please remove the references to the publicly available data from the Data Availability statement, which should only list datasets that were newly generated. Please refer to the re-analysed datasets using 'Data Citations' instead. Here, you cite both, the manuscript that reported these datasets and the dataset itself. Please see the relevant information on Data Citations in our Guide to Authors

<https://www.embopress.org/page/journal/14693178/authorguide#referencesformat>.

You can refer to these datasets either in the text or in the methods, e.g., when you describe public data processing, as you see best fit.

- We have inserted the information on reviewer access in the Data Availability statement for easy reviewer access. Please make sure that you remove it now.

- The title for the Methods section is simply 'Methods'. Please update this in your manuscript.

- The manuscript sections should be in the following order: Title page - Abstract & Keywords - Introduction - Results - Discussion - Methods - Data Availability - Acknowledgments - Disclosure Statement & Competing Interests - References - Figure Legends - (Main Tables with legends) - Expanded View Figure Legends.

- Figure EV4: the scale bars contain text (numbers) which will not be well visible. Please use only lines in the figure panel and define the size exclusively in the figure legend.

- Our production/data editors have asked you to clarify several points in the figure legends (see below). Please incorporate these changes in the manuscript and return the revised file with tracked changes with your final manuscript submission.

A) Statistical test information. Only p-values that are actually shown in the figure panel(s) should (and must) be defined in the

legends, all others should be removed from (or added to) the legend. Moreover, we ask for the specification of exact p-values:

- Please note that the exact p values are not provided in the legends of figures 1d-e, g-i; 2b-f, k-j; 3d, f-i; 4d, f, h, j; 5b, d, f, h-j; 6b, d, f, h, j, l; 7a-f, h-k; 8a-f, h-k; EV 1a-c; EV 4e; EV 5b, d, f, h.
- Please indicate the statistical test used for data analysis in the legends of figures 3a; EV 3b-d.
- Please note that in figures 2b-f, k-j; 3d, f-i; 4d, f, h, j; 5b, d, f, h-j; 6b, d, f, h, j, l; 7a-f, h-k; 8a-f, h-k; EV 4e; there is a mismatch between the annotated p values in the figure legend and the annotated p values in the figure file that should be corrected.

B) Replicates and error bars:

- Please note that the box plots need to be defined in terms of minima, maxima, centre, bounds of box and whiskers, and percentile in the legend of figure EV 1a.
- Please note that information related to n is missing in the legends of figures 1d-i; EV 1a; EV 3b.
- Although 'n' is provided, please describe the nature of entity for 'n' in the legends of figures 7a-f, h-j; 8a-f, h-j.

- As a standard procedure we edit the abstract of manuscripts. Please find my suggestion below my signature

- Finally, EMBO Reports papers are accompanied online by

A) a short (1-2 sentences) summary of the findings and their significance,

B) 2-3 bullet points highlighting key results and

C) a schematic summary figure that provides a sketch of the major findings (not a data image).

Please provide the summary figure as a separate file in PNG or JPG format at a size of 550x300-600 pixels (width x height).

Please note that the size is rather small and that text needs to be readable at the final size. Please send us this information along with the revised manuscript.

- On a different note, I would like to alert you that EMBO Press offers a new format for a video-synopsis of work published with us, which essentially is a short, author-generated film explaining the core findings in hand drawings, and, as we believe, can be very useful to increase visibility of the work. This has proven to offer a nice opportunity for exposure i.p. for the first author(s) of the study. Please see the following link for representative examples and their integration into the article web page:

<https://www.embopress.org/doi/full/10.15252/emj.2019103932>

With kind regards,

Martina Rembold, PhD

Senior Editor

EMBO reports

Referee #1:

1. Overall rating.

Good.

2. Suitability for publication.

Yes after minor revision, see remark to author.

3. Clarity / figures.

Good.

4. Remarks to the Author.

- The authors have adequately addressed my concerns. It would have been helpful if the authors highlighted the modifications made in the new manuscript, to more rapidly review the adaptations.

- Adapt the title of figure 7. As Figure 7 and 8 are exactly the same (except for LPS vs CLP), I would use the same title (but of course adapt it to CLP or LPS).

Figure 7; "Overexpression of TLR4 Reverses the Protective Effect of Cmtm3 KO in LPS-Induced Endotoxemia."

- A concluding figure or graphical abstract would be helpful to effectively summarize the key findings and insights of the current work.

Referee #2:

The authors have nicely addressed my concerns.

Abstract

Regulation of neutrophil activation plays a significant role in managing sepsis. CKLF-like MARVEL transmembrane domain containing (CMTM)3 is a membrane protein involved in immune responses and tumor development. Here, we find that CMTM3 expression is elevated in sepsis and plays a crucial role in mediating the imbalance of neutrophil migration. Cmtm3 knockout improves the survival rate of septic mice, mitigates inflammatory responses, and ameliorates organ damage. Mechanistically, the deletion of Cmtm3 reduces the expression of Toll-like receptor 4 (TLR4) on neutrophils, leading to a decrease in the expression of C-X-C motif chemokine receptor 2 (CXCR2) on the cell membrane. This results in a reduced migration of neutrophils from the bone marrow to the bloodstream, thereby attenuating their recruitment to vital organs. Our findings indicate that targeting CMTM3 holds promise as a valuable therapeutic approach to ameliorate the dysregulation of neutrophil migration and multi-organ damage associated with sepsis.

All editorial and formatting issues were resolved by the authors.

Prof. Fengxue Zhu
Peking University People's Hospital
Department of Critical Care Medicine
11 Xizhimen South Street
Beijing, Beijing 100044
China

Dear Prof. Zhu,

I am very pleased to accept your manuscript for publication in the next available issue of EMBO reports. Thank you for your contribution to our journal.

Yours sincerely,
